

# Atmospheric energy budget response to idealized aerosol perturbation in tropical cloud systems

**Guy Dagan[1], Philip Stier[1], Matthew Christensen[1], Guido Cioni[2,3], Daniel Klocke[3,4] and Axel Seifert[4]**

[1] Atmospheric, Oceanic and Planetary Physics, Department of Physics, University of Oxford, UK

[2] Max Planck Institute for Meteorology, Hamburg, Germany

[3] Hans Ertel Center for Weather Research, Offenbach am Main, Germany

[4] Deutscher Wetterdienst, Offenbach am Main, Germany

E-mail: guy.dagan@physics.ox.ac.uk

## Abstract

The atmospheric energy budget is analysed in numerical simulations of tropical cloud systems. This is done in order to better understand the physical processes behind aerosol effects on the atmospheric energy budget. The simulations include both shallow convective clouds and deep convective tropical clouds over the Atlantic Ocean. Two different sets of simulations, at different dates (10-12/8/2016 and 16-18/8/2016), are being simulated with different dominant cloud modes (shallow or deep). For each case, the cloud droplet number concentrations (CDNC) is varied as a proxy for changes in aerosol concentrations. It is shown that the total column atmospheric radiative cooling is substantially reduced with CDNC in the deep-cloud dominated case (by ~10.0 W/m$^2$), while a much smaller reduction (~1.6 W/m$^2$) is shown in the shallow-cloud dominated case. This trend is caused by an increase in the ice and water vapor content at the upper troposphere that leads to a reduced outgoing longwave radiation. A decrease in sensible heat flux (driven by increase in the near surface air temperature) reduces the warming by ~1.4 W/m$^2$ in both cases. It is also shown that the cloud fraction response behaves in opposite ways to an increase in CDNC, showing an increase in the deep-cloud dominated case and a decrease in the shallow-cloud dominated case. This demonstrates that under different environmental conditions the response to aerosol perturbation could be different.



## **Introduction**

The negative anthropogenic radiative forcing due to aerosols is acting to cool the climate and to
compensate some of the warming due to increase in greenhouse gases (Boucher et al., 2013).
However, quantification of this effect is highly uncertain with a revised uncertainty range of
$-1.60$ to $-0.65$ W/m$^2$ (Bellouin et al., 2019). The total anthropogenic aerosol radiative forcing is
composed of contribution from direct interaction of aerosols with radiation (scattering and
absorption) and from indirect interaction with radiation due to changes in cloud properties.
Beside its effect on the radiation budget, aerosols may affect the precipitation distribution and
total amount (Levin and Cotton, 2009; Albrecht, 1989; Tao et al., 2012). A useful perspective to
improve our understanding of aerosol effect on precipitation, which became common in the last
few years, arises from constraints on the energy budget (O'Gorman et al., 2012; Muller and
O'Gorman, 2011; Hodnebrog et al., 2016; Samset et al., 2016; Myhre et al., 2017; Liu et al.,
2018; Richardson et al., 2018; Dagan et al., 2019a). On long time scales, any precipitation
perturbations by aerosol effects will have to be balanced by changes in radiation fluxes, sensible
heat flux or by divergence of dry static energy. The energy budget constraint perspective was
found useful to explain both global (e.g. (Richardson et al., 2018)) and regional (Liu et al., 2018;
Dagan et al., 2019a) precipitation response to aerosol perturbations in global scale simulations.
In this study, we investigate the energy budget response to aerosol perturbation on a regional
scale using high resolution cloud resolving simulations. This enables an improved understanding
of the microphysical processes controlling atmospheric energy budget perturbations. The strong
connection between the atmospheric energy budget and convection has long been appreciated
(e.g. (Arakawa and Schubert, 1974; Manabe and Strickler, 1964)) as well as the connection to
the general circulation of the atmosphere (Emanuel et al., 1994).
The total column atmospheric energy budget can be described as follows:
$\quad LP + Q_R + Q_{SH} = \text{div}(s) + \text{d}s/\text{dt}$ \qquad (1)
Equation 1 presents a balance between the latent heating rate ($LP$ - latent heat of condensation
[$L$] times the surface precipitation rate [$P$]), the surface sensible heat flux ($Q_{SH}$), the atmospheric
radiative heating ($Q_R$), the divergence of dry static energy (div($s$), which will become negligible
on sufficiently large spatial scales), and dry static energy storage term (d$s$/dt, which will become
negligible on long [inter-annual] temporal scales). Throughout the rest of this paper we will refer
to the right-hand side of Equation 1 (div($s$)+d$s$/dt) as the residual (R) of the left-hand side.



$Q_R$ is defined as:
$Q_R = (F_{SW}^{TOA} - F_{SW}^{SFC}) + (F_{LW}^{TOA} - F_{LW}^{SFC})$      (2)
and represents the rate of net atmospheric diabatic warming due to radiative shortwave (SW) and
longwave (LW) fluxes. It is expressed by the sum of the surface (SFC) and top of the atmosphere
(TOA) fluxes, when all fluxes are positive downwards. As in the case of TOA radiative forcing,
aerosols could modify the atmospheric energy budget by both direct interaction with radiation
and by microphysical effects on clouds. The latter is the focus of this study.
The microphysical effects are driven by the fact that aerosols serve as cloud condensation nuclei
(CCN) and ice nuclei (IN). Larger aerosol concentrations, e.g. by anthropogenic emissions, could
lead to larger cloud droplet and ice particle concentrations (Andreae et al., 2004; Twomey, 1977;
Hoose and Möhler, 2012). Changes in hydrometer concentration and size distribution were
shown to affect clouds' microphysical processes rates (such as condensation, evaporation,
freezing and collision-coalescence), which in turn could affect the dynamics of the clouds (Khain
et al., 2005; Koren et al., 2005; Heikenfeld et al., 2019; Chen et al., 2017;Altaratz et al., 2014;
Seifert and Beheng, 2006a), the rain production (Levin and Cotton, 2009; Albrecht, 1989; Tao
et al., 2012) and the clouds' radiative effect (Koren et al., 2010; Storelvmo et al., 2011; Twomey,
1977; Albrecht, 1989). The aerosol effect, and in particular its effects on the radiation budget
and the atmospheric energy budget, is cloud regime dependent (Altaratz et al., 2014; Lee et al.,
2009; Mülmenstädt and Feingold, 2018; van den Heever et al., 2011; Rosenfeld et al., 2013;
Glassmeier and Lohmann, 2016; Gryspeerdt and Stier, 2012; Christensen et al., 2016), time
dependent (Dagan et al., 2017; Gryspeerdt et al., 2015; Seifert et al., 2015; Lee et al., 2012;
Dagan et al., 2018c), aerosol type and size distribution dependent (Jiang et al., 2018; Lohmann
and Hoose, 2009) and (even for a given cloud regime) meteorological conditions dependent
(Dagan et al., 2015a; Fan et al., 2009; Fan et al., 2007; Kalina et al., 2014; Khain et al., 2008)
and was shown to be non-monotonic (Dagan et al., 2015b;J eon et al., 2018; Gryspeerdt et al.,
2019; Liu et al., 2019). Hence the quantification of the global mean radiative effect is extremely
challenging (e.g. (Stevens and Feingold, 2009; Bellouin et al., 2019)).
Previous studies demonstrated that the mean aerosol effect on deep convective clouds can
increase the upward motion of water, and hence also increase the cloud anvil mass and extent
(Fan et al., 2010; Chen et al., 2017; Fan et al., 2013; Grabowski and Morrison, 2016). The
increase in mass flux to upper levels was explained by the convective invigoration hypothesis
(Fan et al., 2013;Koren et al., 2005;  Rosenfeld et al., 2008;Seifert and Beheng, 2006a;  Yuan et



al., 2011a; Williams et al., 2002), which was proposed to lead to stronger latent heat release
under higher aerosol concentrations and hence stronger vertical velocities. In addition to the
stronger vertical velocities, under polluted conditions the smaller hydrometers are being
transported higher in the atmosphere (for a given vertical velocity (Chen et al., 2017; Koren et
al., 2015; Dagan et al., 2018a)) and their lifetime at the upper troposphere is longer (Fan et al.,
2013; Grabowski and Morrison, 2016). The invigoration mechanism can also lead to an increase
in precipitation (Khain, 2009; Altaratz et al., 2014). Both the increase in precipitation and the
increase in anvil coverage would act to warm the atmospheric column: the increased precipitation
by latent heat release, and the increased anvil mass and extent by longwave radiative warming
(Koren et al., 2010; Storelvmo et al., 2011). However, it should be pointed out that the
uncertainty underlying these proposed effects remain significant (White et al., 2017; Varble,
2018). In addition, aerosol effects on precipitation from deep convective cloud was shown to be
non-monotonic and depend on the aerosol range (Liu et al., 2019).
In the case of shallow clouds, aerosol effect on precipitation was also shown to be non-monotonic
(Dagan et al., 2015a;Dagan et al., 2017). However, unlike in the deep clouds case, the mean
effect on precipitation, under typical modern-day conditions, is thought to be negative (Albrecht,
1989; Rosenfeld, 2000; Jiang et al., 2006; Xue and Feingold, 2006; Dagan and Chemke, 2016).
The aerosol effect on shallow cloud cover and mean water mass (measure by liquid water path -
LWP) might also depend on the meteorological conditions and aerosol range (Dagan et al.,
2015b;  Dagan et al., 2017; Gryspeerdt et al., 2019; Dey et al., 2011; Savane et al., 2015) and is
the outcome of competition between different opposing response of: rain suppression (that could
lead to increase in cloud lifetime and coverage (Albrecht, 1989)), warm clouds invigoration (that
could also lead to increase in cloud coverage and LWP (Koren et al., 2014; Kaufman et al., 2005;
Yuan et al., 2011b)) and increase in entrainment and evaporation (that could lead to decrease in
cloud coverage (Small et al., 2009;  Jiang et al., 2006;Costantino and Bréon, 2013; Seigel, 2014)).
Another addition to this complex response is the fact that the aerosol effect on warm convective
clouds was shown to be time dependent and affected by the clouds' feedbacks on the
thermodynamic conditions (Seifert et al., 2015; Dagan et al., 2016; Dagan et al., 2017; Lee et al.,
2012; Stevens and Feingold, 2009; Dagan et al., 2018b). Previous simulations that contained
several tropical cloud modes demonstrate that increase in aerosol concentrations can lead to
suppression of the shallow mode and invigoration of the deep mode (van den Heever et al., 2011).
Hence the domain mean effect, even if it is demonstrated to be small, may be the result of
opposing relatively large contributions from the different cloud modes (van den Heever et al.,



2011). The small domain mean effect may suggest that on large enough scales the energy (Muller
and O'Gorman, 2011; Myhre et al., 2017) or water budget (Dagan et al., 2019b) constrain
precipitation changes.
Previous studies, using global simulations (O'Gorman et al., 2012; Muller and O'Gorman, 2011;
Hodnebrog et al., 2016; Samset et al., 2016; Myhre et al., 2017; Liu et al., 2018; Richardson et
al., 2018;Dagan et al., 2019a), demonstrated the usefulness of the atmospheric energy budget
perspective in constraining aerosol effect on precipitation. However, the physical processes
behind aerosol-cloud microphysical effects on the energy budget are still far from being fully
understood. In this study we use cloud resolving simulations to increase our understanding of the
effect of microphysical aerosol-cloud interactions on the atmospheric energy budget.
**Methodology**
The icosahedral nonhydrostatic (ICON) atmospheric model (Zängl et al., 2015) is used in a
limited area configuration. ICON's non-hydrostatic dynamical core was evaluated with several
idealized cases (Zängl et al., 2015). The simulations are conducted such that they are aligned
with the NARVAL 2 (Next-generation Aircraft Remote-Sensing for Validation Studies (Klepp
et al., 2014;S tevens et al., 2019; Stevens et al., 2016)) campaign, which took place during August
2016 in the western part of the northern tropical Atlantic. We use existing NARVAL 2
convection-permitting simulations (Klocke et al., 2017) as initial and boundary conditions for
our simulations.
The domain covers ~22$^\circ$ in the zonal direction (25$^\circ$ - 47$^\circ$ W) and ~11$^\circ$ in the meridional direction
(6$^\circ$ - 17$^\circ$ N) and therefore a large fraction of the northern tropical Atlantic (Fig. 1). During August
2016, the intertropical convergence zone (ITCZ) was located in the southern part of the domain
while the northern part mostly contains trade cumulus clouds. Hence, this case study provides
an opportunity to study heterogenous clouds systems. Daily variations in the deep/shallow cloud
modes in our domain were observed, but it always included both cloud modes, albeit in different
relative fraction. Two different dates are chosen, one representing a shallow-cloud dominated
mode (10-12/8/2016 – see Fig. 2, and Figs S1 and S3, supporting information- SI), and one that
represents a deep-cloud dominated mode (16-18/8/16 – see Fig. 3 and Figs. S2 and S3, SI). In
the shallow-cloud dominated case, most of the domain is covered by trade cumulus clouds that
are being advected with the trade winds from north-east to south-west. In the southern part of the
domain, throughout most of the simulation, there is a zonal band of deep convective clouds (Fig.



2) that contribute on average ~25% out of the total cloud cover (Fig. S3, SI). The deep-cloud
dominated case represents the early stages of the development of the tropical storm Fiona (Fig.
3). Fiona formed in the eastern tropical Atlantic and moved toward the west-north-west. It started
as a tropical depression at 16/8/2016 18:00 UTC while its centre was located at 12.0° N 32.2° W.
It kept moving towards the north-west and reach a level of a tropical storm at 17/8/2016 12UTC,
while its centre was located at 13.7° N 36.0° W
(https://www.nhc.noaa.gov/data/tcr/AL062016_Fiona.pdf). The general propagation speed and
direction, strength (measure by maximal surface wind speed) and location of the storm are
predicted well by the model. However, the model produces more anvil clouds than what was
observed from the satellite (Fig. 3). These two different cases, representing different atmospheric
energy budget initial state (see also Figs. 4 and 12 below), enable the investigation of the aerosol
effect on the energy budget under different initial conditions.
We use a two-moment bulk microphysical scheme (Seifert and Beheng, 2006b). For each case,
four different simulations with different prescribed cloud droplet number concentrations
(CDNC) of 20, 100, 200, and 500 cm$^{-3}$ are conducted. The different CDNC scenarios serve as
a proxy for different aerosol concentration conditions (as the first order effect of increased
aerosol concentration is to increase the CDNC (Andreae, 2009)) and avoid the uncertainties
involved in the representation of the aerosols in numerical models (Ghan et al., 2011; Simpson
et al., 2014; Rothenberg et al., 2018). However, it limits potential feedbacks between clouds
and aerosols, such as the removal of aerosol levels by precipitation scavenging and potential
aerosol effects thereon. In addition, the fix CDNC framework does not capture the differences
in aerosol activation fraction between shallow and deep clouds, due to differences in vertical
velocity.
For calculation of the difference between high CDNC (polluted) conditions and low CDNC
(clean) conditions, the simulations with CDNC of 200 and 20 cm$^{-3}$ are chosen as they represent
the range typically observed over the ocean (see for example the CDNC range presented in
recent observational-based studies (Rosenfeld et al., 2019; Gryspeerdt et al., 2019)). Each
simulation is conducted for 48 hours starting from 12 UTC. The horizontal resolution is set to
1200 m and 75 vertical levels are used. The temporal resolution is 12 sec and the output interval
is 30 min. Interactive radiation is calculated every 12 min using the RRTM-G scheme (Clough
et al., 2005; Iacono et al., 2008; Mlawer et al., 1997). We have added a coupling between the
microphysics and the radiation to include the Twomey effect (Twomey, 1977). This was done



by including the information of the cloud liquid droplet effective radius, calculated in the
microphysical scheme, in the radiation calculations. No Twomey effect due to changes in the
ice particles size distribution was considered due to the large uncertainty involved in the ice
microphysics and morphology. Additional details, such as the surface and atmospheric physics
parameterizations, are described in Klocke et al., (2017) and include an interactive surface flux
scheme and a fixed sea surface temperature.
For comparing the outgoing longwave flux from the simulations and observations we use
imager data from the SEVIRI instrument onboard the Meteosat Second Generation (MSG)
geostationary satellite (Aminou, 2002). The outgoing longwave flux is calculated using the
Optimal Retrieval for Aerosol and Cloud (ORAC) algorithm (Sus et al. 2017; McGarragh, et
al. 2017). Cloud optical (thickness, effective radius, water path) and thermal (cloud top
temperature and pressure) properties are retrieved from ORAC using an optimal estimation-
based approach. These retrievals and reanalysis profiles of temperature, humidity and ozone
are then ingested into BUGSrad, a two-stream correlated-k broadband flux algorithm (Stephens
et al., 2001) that outputs the fluxes at the top and bottom of the atmosphere and shown to have
excellent agreement when applied to both active (CloudSat) and passive (Advanced Along
Track Scanning Radiometer) satellite sensors compared to Clouds and the Earth's Radiant
Energy System (Henderson et al. 2013; Stengel et al. 2019). In addition, off-line sensitivity
radiative transfer tests using vertical profiles from our model were conducted with BUGSrad
to identify the source of the differences in fluxes between clean and polluted conditions.




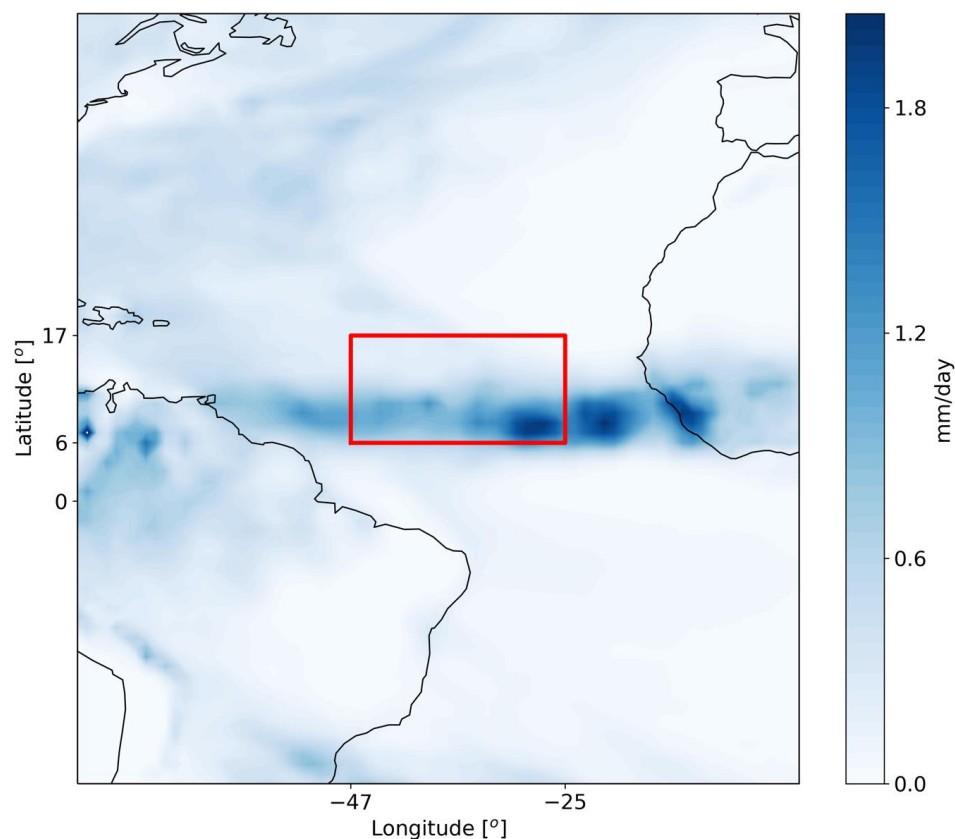

**Figure 1. Domain of the ICON simulations (red rectangle) for the NARVAL 2 case study overlaid on the**
**August 2016 ECMWF era-interim reanalysis (Dee et al., 2011) mean precipitation rate.**





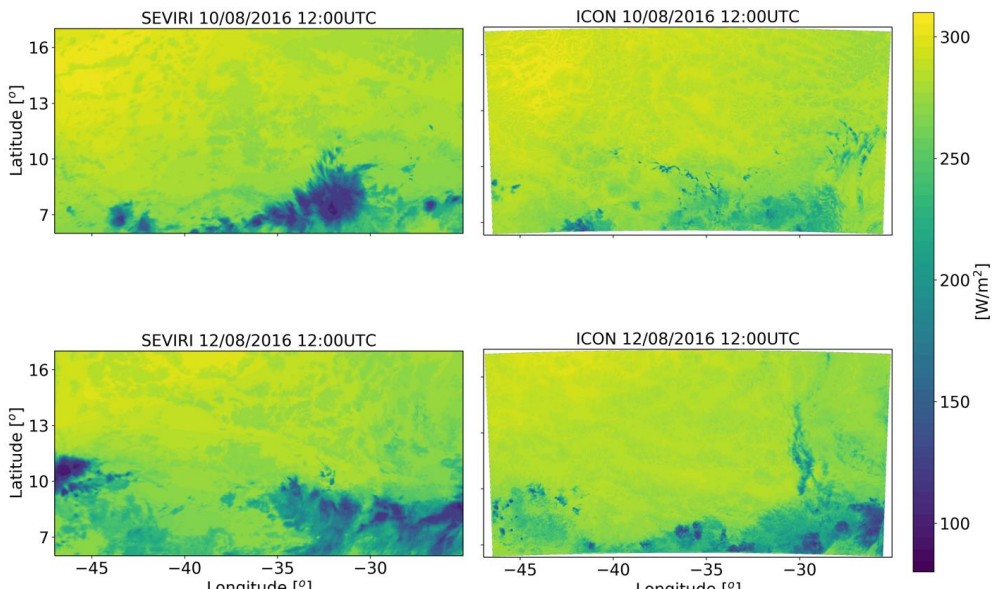

**Figure 2. Outgoing longwave flux at the top of atmosphere at the initial stage (upper row) and the last stage (lower row – each average over 30 minutes) of the simulation of the shallow-cloud dominated case (10-12/08/2016) from geo-stationary satellite (SEVIRI-MSG – right column) and the ICON model simulation with CDNC of 20 cm$^{-3}$ (left column).**

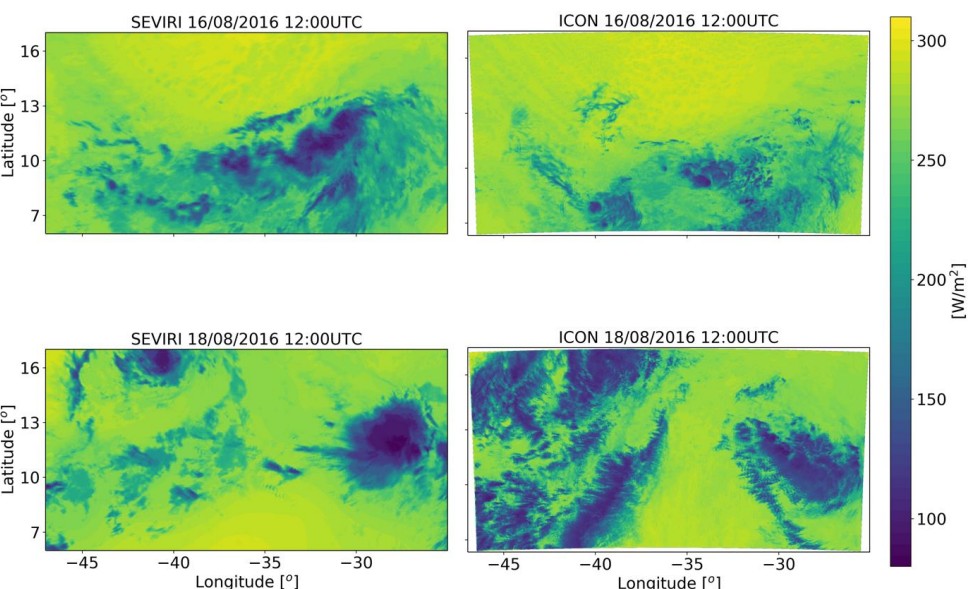

**Figure 3. similar to Figure 2 but for the deep-cloud dominated case.**
**Results**
**Shallow-cloud dominated case -10-12/08/2016**
We start with energy budget analysis of the shallow-cloud dominated case base simulations
(CDNC = 20 cm$^{-3}$). Figure 4 presents the time mean (over the two days simulation) of the
different terms of the energy budget (Equation 1). As expected, $LP$ dominates the warming of
the atmosphere while $Q_R$ dominate the cooling. The sensible heat flux ($Q_{SH}$) is positive (act to
warm the atmosphere) but it is an order of magnitude smaller than the $LP$ and $Q_R$ magnitudes. In
this shallow-cloud dominated case the radiative cooling of the atmosphere is significantly larger
than the warming due to precipitation (mean of -114.7 W/m$^2$ compare with 90.1 W/m$^2$), hence
the residual (R) is negative. Negative R means that there must be some convergence of dry static
energy into the domain and/or decrease in the storage term.
We note that there is a significant difference in the spatial distribution of $LP$ and $Q_R$ (Jakob et
al., 2019). While the $Q_R$ is more uniformly distributed, the $LP$ is mostly concentrated at the south
part of the domain (where the deep convective clouds are formed) and it has a dotted structure.
Locally, at the core of a deep convective clouds, the $LP$ contribution can reach a few 1000 W/m$^2$
(1 mm/hr of precipitation is equivalent to 628 W/m$^2$), however, the vast majority of the domain
contributes very little in terms of $LP$. $Q_R$ also presents some spatial structure in which there is a



weak atmospheric cooling at the south part of the domain (the region of the deep convective
clouds) and a strong cooling at the reset of the domain.

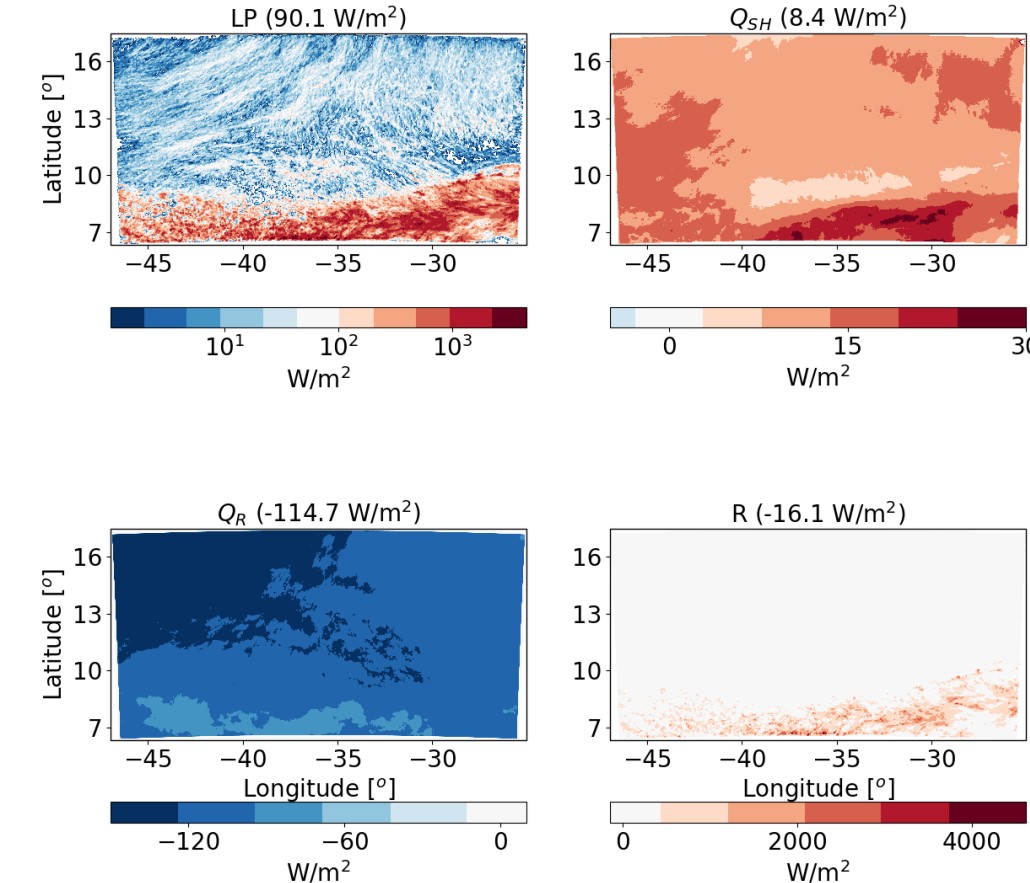


**Figure 4. Spatial distribution of the time mean of the different terms of the energy budget for the ICON**

**simulation of the shallow-cloud dominated case (10-12/08/2016) with CDNC = 20 cm$^{-3}$. The terms that appear**

**here are: $LP$ - latent heat by precipitation, $Q_{SH}$ - sensible heat flux, $Q_R$ - atmospheric radiative warming, and**

**$R$ – the residual. The domain and time-mean value of each term appears in parenthesis.**


For understanding the spatial structure of $Q_R$, next we examine the spatial distribution of the LW
and SW radiative fluxes at the TOA and surface (Fig. 5). We note that the smaller radiative
cooling in the region of deep clouds in the south of the domain is mostly contributed by a
decrease in $F_{LW}^{TOA}$. The SW fluxes also demonstrate a strong south-north gradient, as the deep





convective clouds in the south are more reflective than the shallow trade cumulus (with the lower
mean cloud fraction) in the rest of the domain.

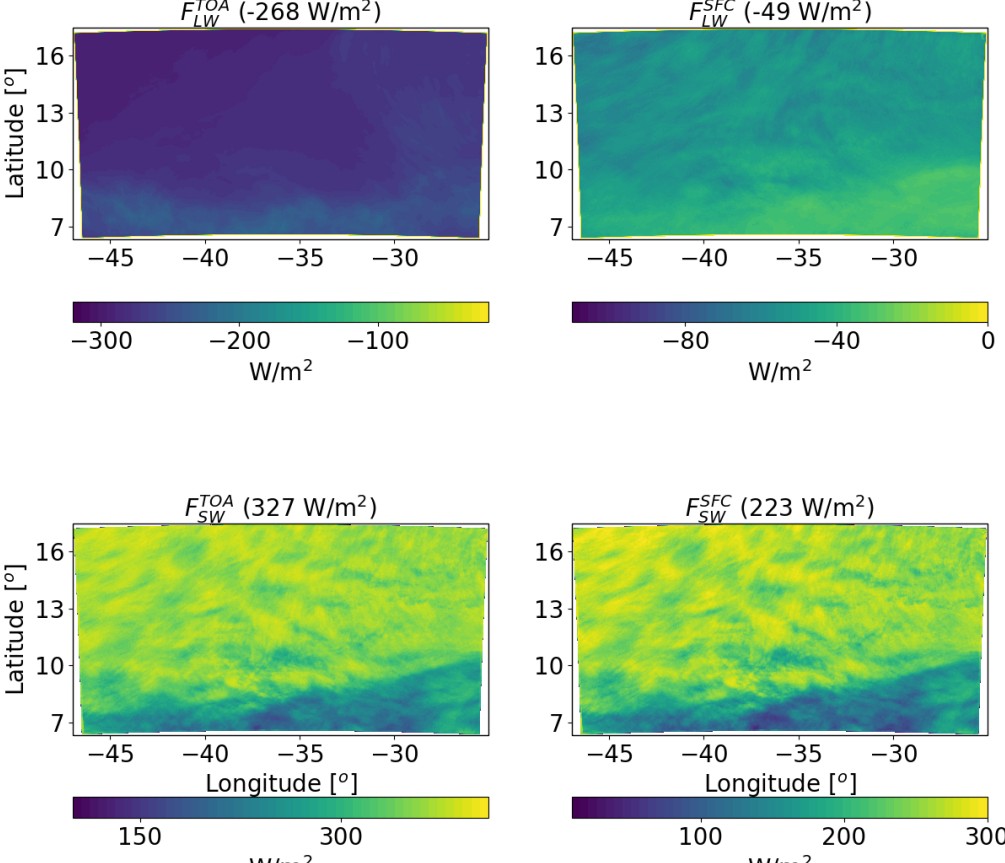


**Figure 5. Spatial distribution of ICON simulated time-mean longwave (LW) and shortwave (SW) radiation**
**fluxes at the top of atmosphere (TOA) and surface (SFC) for a simulation of the shallow-cloud dominated**
**case (10-12/08/2016) with CDNC = 20 cm$^{-3}$. The domain and time mean value of each term appears in**
**parenthesis.**

**Response to aerosol perturbation – shallow-cloud dominated case**
Next, we analyse the response of the atmospheric energy budget of this case to perturbations in
CDNC. Figure 6 presents the differences in the different terms of the energy budget between a





polluted simulation (CDNC = 200 cm$^{-3}$) and a clean simulation (CDNC = 20 cm$^{-3}$). It
demonstrates that the *LP* differences between the different CDNC scenarios contribute 5.1 W/m$^2$
less to warm the atmosphere in the polluted vs. the clean simulation. We note that this apparently
large effect is caused by a small, non-statistically significant, precipitation difference (~0.4 mm
over the two days of simulation - see Fig. 8 below). The strong sensitivity of the atmospheric
energy budget to small precipitation changes (recalling that 1 mm/hr is equivalent to 628 W/m$^2$)
exemplifies the caution one needs to take when looking on precipitation response in terms of
energy budget perspective. The $Q_R$ differences lead to relative warming of the atmosphere of the
polluted case compared to the clean case by 1.6 W/m$^2$. We note that most of the $Q_R$ differences
are located in the south-west part of the domain. The $Q_{SH}$ changes counteracts 1.4 W/m$^2$ of the
atmospheric warming by $Q_R$ and so the end result is a deficit of 4.8 W/m$^2$ in the atmospheric
energy budget in the polluted simulation compared to the clean simulation. The decrease in the
$Q_{SH}$ is driven by an increase in the near surface air temperature (see Fig. 8).



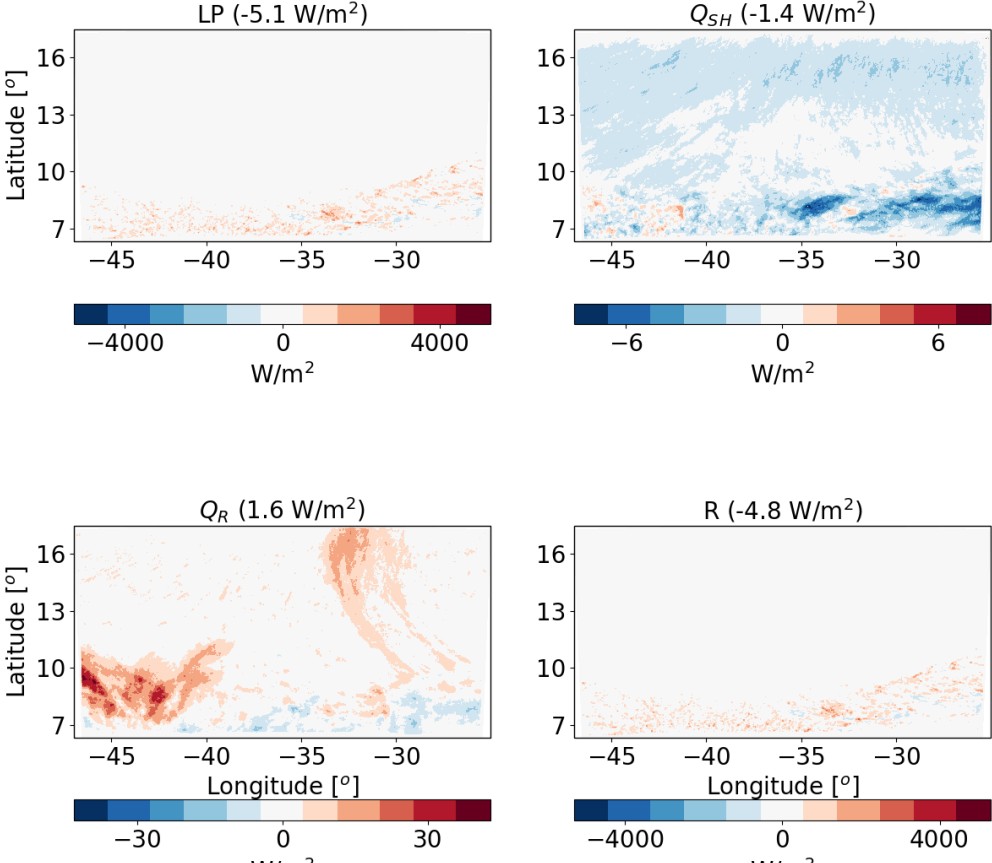


**Figure 6. The differences between polluted (CDNC = 200 cm$^{-3}$) and clean (CDNC = 20 cm$^{-3}$) ICON simulations**
**of the time-mean terms of the energy budget for the shallow-cloud dominated case (10-12/08/2016). The terms**
**that appears here are: *LP* - latent heat by precipitation, $Q_{SH}$ - sensible heat flux, $Q_R$ - atmospheric radiative**
**warming, and R – the residual. The domain and time mean value of each term appears in parenthesis.**

To understand the response of $Q_R$ to the CDNC perturbation, we next examine the response of
the different radiative fluxes. Figure 7 demonstrates that most of the relative atmospheric
radiative heating in the polluted case compared to the clean case is contributed by changes in the
$F_{LW}^{TOA}$ fluxes. The changes in $F_{LW}^{SFC}$ are an order of magnitude smaller. The SW fluxes change both
at the TOA and SFC are larger than the $F_{LW}^{TOA}$ changes, however, in terms of the atmospheric energy
budget, they almost cancel each other out and the net SW atmospheric effect is only -0.9 W/m$^2$.
Most of the reduction in SW fluxes (both at TOA and the surface) comes from the deep
convective regions in the south of the domain while the shallow cloud regions experience some
increase in SW fluxes. This can be attributed to the increase in deep convective cloud fraction





and a decrease in the shallow cloud fraction with the increase in CDNC (see Fig. 9 below). The
TOA net radiative effect for the entire system (as opposed to the atmospheric energy budget that
take into consideration the surface radiative fluxes changes) is about -5.2 W/m$^2$.

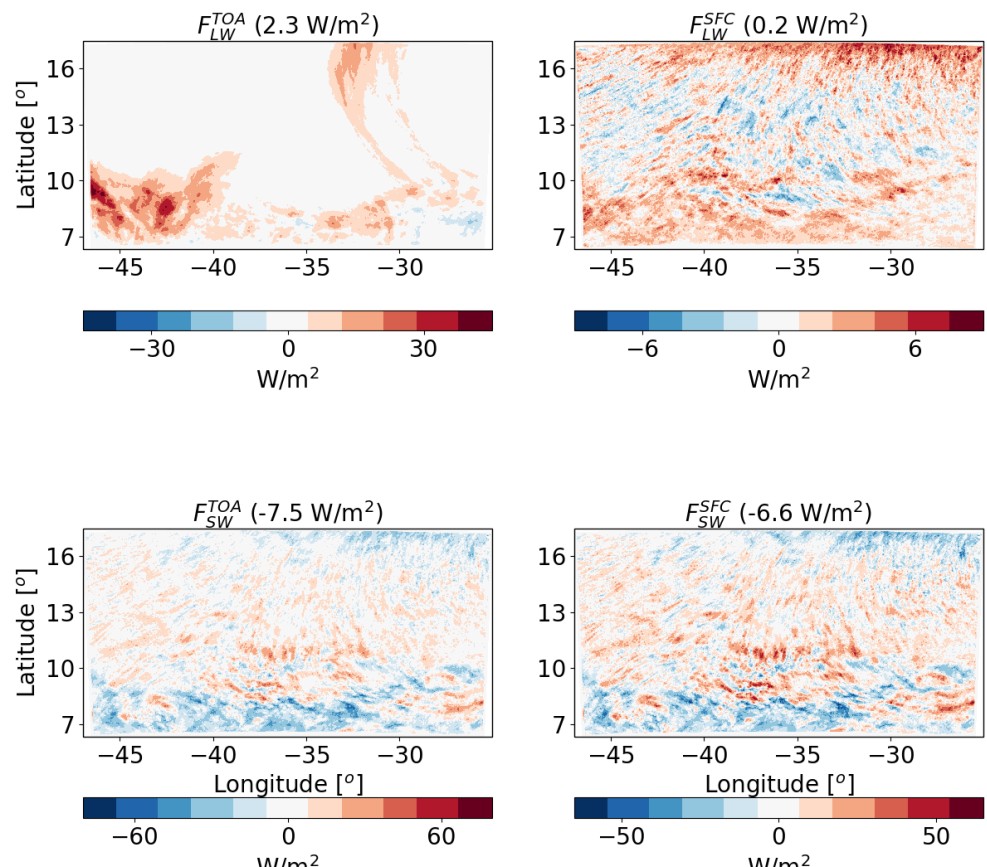


**Figure 7. The differences between polluted (CDNC = 200 cm$^{-3}$) and clean (CDNC = 20 cm$^{-3}$) ICON simulations**
**of the time mean radiative longwave (LW) and shortwave (SW) fluxes at the top of atmosphere (TOA) and**
**surface (SFC) for the shallow-cloud dominated case (10-12/08/2016). The domain and time mean value of**
**each term appears in parenthesis.**

The differences in the energy (Fig. 6) and radiation (Fig. 7) budgets between the clean and
polluted cases shown above, could be explained by the differences in the cloud mean properties.
Figure 8 presents the time evolution of some of the domain mean properties while Fig. 9 presents
time and horizontal mean vertical profiles. To examine the robustness of the trends we add here
two more CDNC cases of 100 and 500 cm$^{-3}$ (on top of the two that were examine above – 20 and



305 200 cm$^{-3}$). Figure 8 demonstrates that the domain mean cloud fraction (CF) generally decreases

306 with the increase in CDNC (except for the first ~10 hours of the simulations). Examining the

307 vertical structure of the CF response (Fig. 9), demonstrates that with the increase in CDNC there

308 is a reduction in the low level (below 800 mb) CF concomitantly with an increase in CF at the

309 middle and upper troposphere. The differences in rain rate between the different simulations are

310 small. However, both the liquid water path (LWP) and the ice water path (IWP) show a consistent

311 increase with CDNC. Accordingly, also the total water path (TWP), which is the sum of the LWP

312 and the IWP, substantial increases with CDNC. The vertical profiles of the different hydrometers

313 (Fig. 9) indicate, as expected, that the cloud droplet mass mixing ration ($q_c$ - droplet with radius

314 smaller than 40 μm) increases with CDNC, while the rain mass mixing ratio ($q_r$ - drops with

315 radius larger than 40 μm) decreases due to the shift in the droplet size distribution to smaller

316 sizes under larger CDNC conditions. As this case is dominated by shallow clouds, there exists

317 only a comparably small amount of ice mixing ration ($q_i$) (c.f. Fig. 17), but its concentration

318 increases with the CDNC increase. The combined effect of the increase in CDNC is to

319 monotonically increase the total water mixing ratio ($q_t$) above 800 mb (Fig. 9). The relative

320 increase in $q_t$ with CDNC becomes larger at higher levels.

321 The increase in cloud water with increasing CDNC can explain both the reductions in the net

322 downward SW fluxes (both at TOA and surface) and the decrease in outgoing LW flux at TOA

323 (Fig. 7), as it results in more SW reflection concomitantly with more LW trapping in the

324 atmosphere (Koren et al., 2010). Another contributor to the SW flux reduction (more reflectance)

325 at the TOA is the Twomey effect (Twomey, 1977), while, the decrease in the low-level CF

326 compensates some of this effect. Here we present the outcome of these contradicting effects on

327 the SW fluxes, which shows a reduction at both the TOA and surface (Fig. 7). For estimating the

328 relative contribution of the Twomey effect compare to the cloud adjustments (CF and TWP

329 effects) to the SW flux changes, we have re-run the simulations with the Twomey effect turned

330 off (the radiation calculations do not consider the changes in effective radius between the

331 different simulations). It demonstrates that without the Twomey effect the TOA SW difference

332 is only -1.7 W/m$^2$ as compared to -7.5 W/m$^2$ with the Twomey effect, demonstrating the

333 predominant role of the Twomey effect. For estimating the relative contribution of the changes

334 in CF and TWP to the SW flux changes we have conducted off-line radiative transfer sensitivity

335 tests. To quantify the TWP radiative effect we feed the same CF vertical profile from the model

336 into BUGSrad while allowing the TWP vertical profile to change (and visa versa to compute the





CF radiative effect). This approach demonstrates that the contribution from the small reduction
in CF is negligible compared to the increased SW reflectance caused by the increase in TWP.
We also note a monotonic increase in the near surface temperature with CDNC (see also Fig. 10
below). This trend can be explained by warm rain suppression with increasing CDNC that leads
to less evaporative cooling (see the decrease in the total amount of water mass mixing ration just
above the surface in Fig. 9, (Dagan et al., 2016; Albrecht, 1993; Seigel, 2014; Seifert and Heus,
2013; Lebo and Morrison, 2014)). In addition, it was shown that under polluted conditions the
rain drops below cloud base are larger, hence evaporating less efficiently (Lebo and Morrison,
2014; Dagan et al., 2016). The increase in the near surface temperature drives the decrease in the
$Q_{SH}$ (Fig. 6).

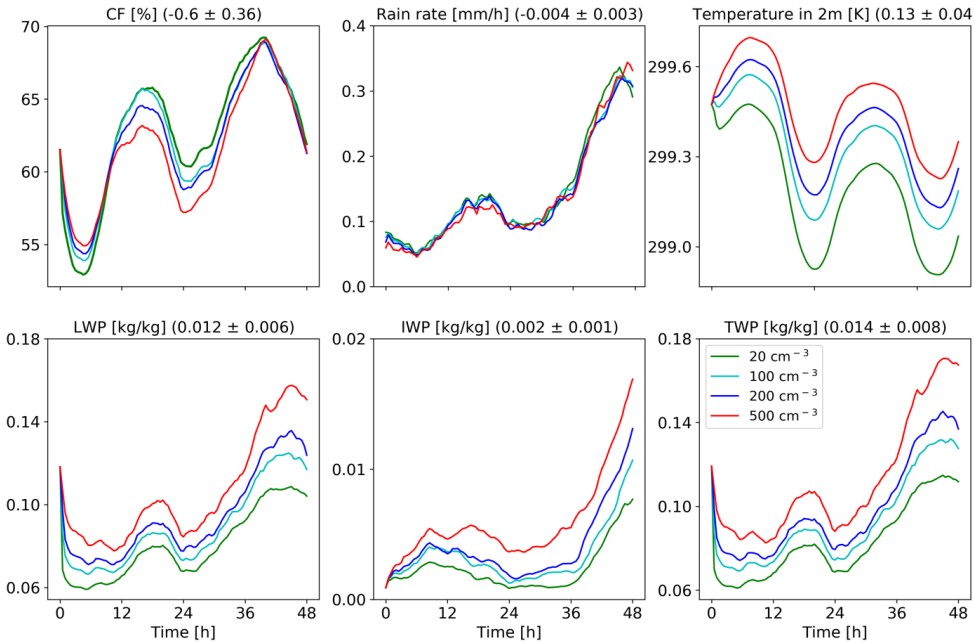


**Figure 8. Domain average properties as a function of time for the different CDNC simulations for the shallow-**
**cloud dominated case. The properties that are presented here are: cloud fraction (CF), rain rate, temperature**
**in 2 m, liquid water path (LWP), ice water path (IWP) and total water path (TPW = LWP + IWP). For each**
**property the mean difference between all combinations of simulations, normalized to a factor 5 increase in**
**CDNC, and its standard deviation appear in parenthesis.**



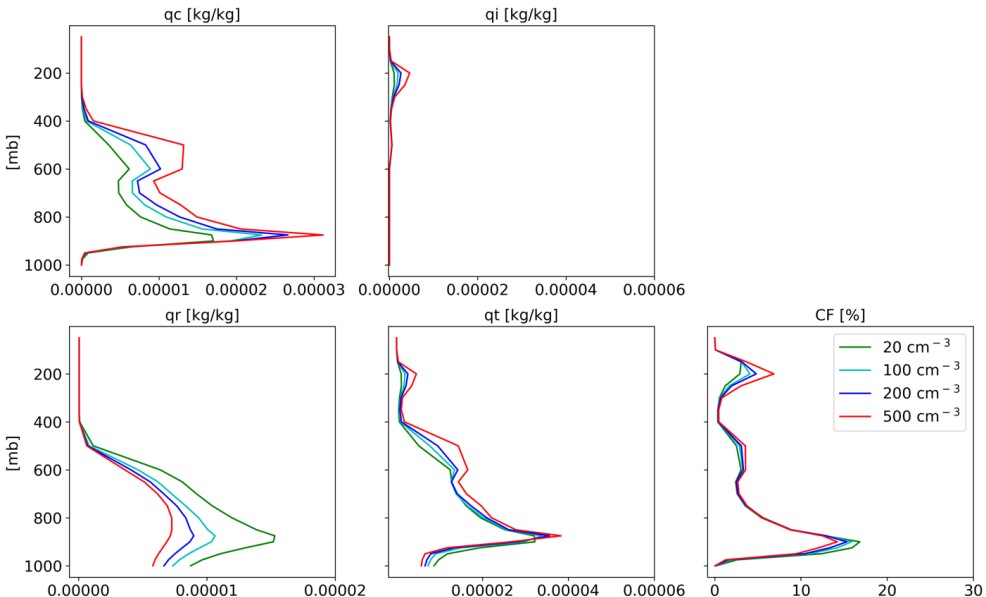


**Figure 9. Domain and time average vertical profiles for the different CDNC simulations for the shallow-cloud**

**dominated case. The properties that are presented here are: cloud droplet mass mixing ratio (qc – for clouds'**

**droplets with radius smaller than 40 μm), ice mass mixing ratio (qi), rain mass mixing ratio (qr - for clouds'**

**drops with radius larger than 40 μm), total water mass mixing ratio (qt = qc+qi+qr), and cloud fraction (CF).**

**The x-axis ranges are identical as for the deep-cloud dominated case – Fig. 17.**


In addition to the clouds' effect on the radiation fluxes, changes in humidity could also contribute
(Fig. 10). We note that increase in CDNC leads to increase in relative humidity (RH) and specific
humidity (qv) at the middle and upper troposphere without a significant temperature change. The
increased humidity at the upper troposphere would act to decrease the outgoing LW flux, a
similar effect as the increased ice content at the upper troposphere has (Fig. 9). However,
sensitivity studies with off-line radiative transfer calculations using BUGSrad, demonstrate that
the vast majority of the different in $F_{LW}^{TOA}$ between clean and polluted conditions emerges from
the cloudy skies (rather than clear-sky), suggesting that the effect of the increased ice content at
the upper troposphere predominant.
Both the increase in water vapor and ice content at the upper troposphere are driven by increase
in water (liquid and ice) mass flux with increasing CDNC to these levels (Fig. 11). The increase
in mass flux is driven partially by the small increase in vertical velocity (especially for updraft





between 5 and 10 m/s) and mostly due to the larger water mass mixing ratio (Fig. 9) that leads
to an increase in mass flux even for a given vertical velocity. The increased relative humidity at
the upper troposphere, further increases the ice particle lifetime at these levels (in addition to the
microphysical effect (Grabowski and Morrison, 2016)) as the evaporation rate decreases. In
addition, the differences in the thermodynamics evolution between the different simulations (Fig.
10) demonstrate drying and warming of the boundary layer with increasing CDNC, due to
reduction in rain evaporation below cloud base and deepening of the boundary layer (Dagan et
al., 2016; Lebo and Morrison, 2014; Seifert et al., 2015; Spill et al., 2019). The drying of the
boundary layer could explain the reduction in the low cloud fraction (Fig. 9 (Seifert et al., 2015)).






**Figure 10. Hovmöller diagrams of the differences in the domain mean temperature, specific humidity (qv)**
**and relative humidity (RH) vertical profiles between polluted (CDNC = 200 cm[-3]) and clean (CDNC = 20 cm[-3])**
**[-3]) simulations for the shallow-cloud dominated case (10-12/08/2016).**

**Figure 11. histograms of ICON simulated vertical velocity at the level of 500 mb (upper), and the time**
**evolution of the net upwards water (liquid and ice) mass flux (lower) for a clean (CDNC = 20 cm[-3]) and**





**polluted (CDNC = 200 cm$^{-3}$) simulations for the shallow-cloud dominated case (10-12/08/2016). The 500 mb**
**level is chosen as it represents the transition between the warm part to the cold part of the clouds.**

### Deep-cloud dominated case -16-18/08/2016

Next, we analyse the atmospheric energy budget for the deep-cloud dominated case (Fiona
tropical storm – Fig. 12). As opposed to the shallow-cloud dominated case, in this case the $LP$
contribution dominates over the radiative cooling and hence the residual R is positive and large.
This difference in the base line atmospheric energy budget between the different cases simulated
here, enable an examination of the aerosol effect on the atmospheric energy budget under
contrasting initial conditions. As in the shallow-cloud dominated case, the $Q_R$ values varies
between small values (especially at the regions that were mostly covered by deep clouds) to
larger negative values (dominated at the regions that were coved by shallow clouds). The $Q_{SH}$ is
positive and an order of magnitude smaller than the $Q_R$ and $LP$, similar to the shallow-cloud
dominated case.

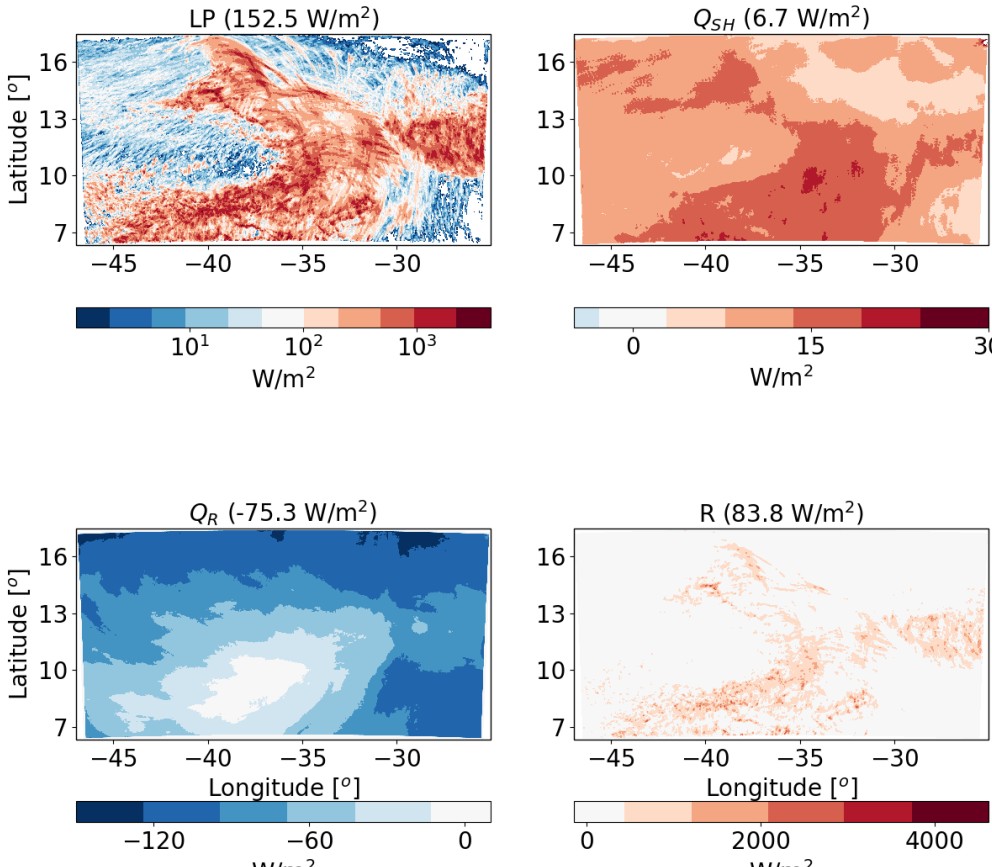

**Figure 12. Spatial distribution of the time mean of the different terms of the energy budget for the ICON simulation of the deep-cloud dominated case (16-18/08/2016) with CDNC = 20 cm$^{-3}$. The terms that appear here are: $LP$ - latent heat by precipitation, $Q_{SH}$ - sensible heat flux, $Q_R$ - atmospheric radiative warming, and R – the residual. The domain and time-mean value of each term appears in parenthesis.**

Further examination of the radiative fluxes (Fig. 13) demonstrates again the resemblance in the spatial structure between $Q_R$ and $F_{LW}^{TOA}$. As compared to the shallow-cloud dominated case, since the clouds are more opaque and cover larger fraction of the sky, there is a decrease in the magnitude of all fluxes (in different amount). For example, $F_{SW}^{SFC}$ is lower by 41 W/m$^2$ (representing larger SW reflectance back to space) and the magnitude of $F_{LW}^{TOA}$ by 47 W/m$^2$ as compare to the shallow-cloud dominated case. The combined effect of the radiative flux





differences between the two cases is a decrease of the atmospheric radiative cooling by 39.6
W/m$^2$ (-114.7 compare with -75.3 W/m$^2$ – see Figs. 5 and 13).

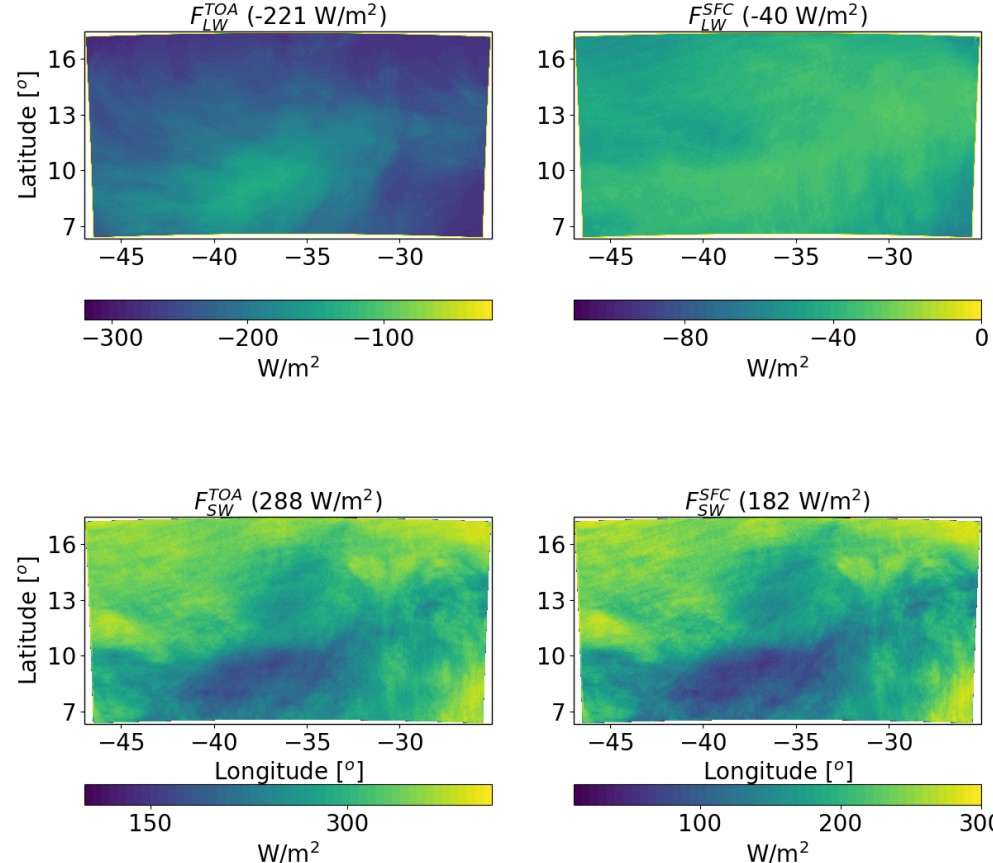


**Figure 13. Spatial distribution of ICON simulated time-mean longwave (LW) and shortwave (SW) radiation**
**fluxes at the top of atmosphere (TOA) and surface (SFC) for a simulation of the deep-cloud dominated case**
**(16-18/08/2016) with CDNC = 20 cm$^{-3}$. The domain and time mean value of each term appears in parenthesis.**

### Response to aerosol perturbation – deep-cloud dominated case

For the deep-cloud dominated case, an increase in CDNC results in a decrease in *LP* by -0.3
W/m$^2$. Again, this difference is due to non-statistically significant precipitation changes (see also
Fig. 16 below). A similar $Q_{SH}$ decrease as in the shallow-cloud dominated case is observed in the
deep-clouds dominated case (see Figs. 14 and 6). The predominant difference in the response



between the two cases is in $Q_R$, which increases much more in the deep-cloud dominated case:
10.0 W/m$^2$ (Fig. 14) compared with 1.6 W/m$^2$ in the shallow-cloud dominated case (Fig. 6).

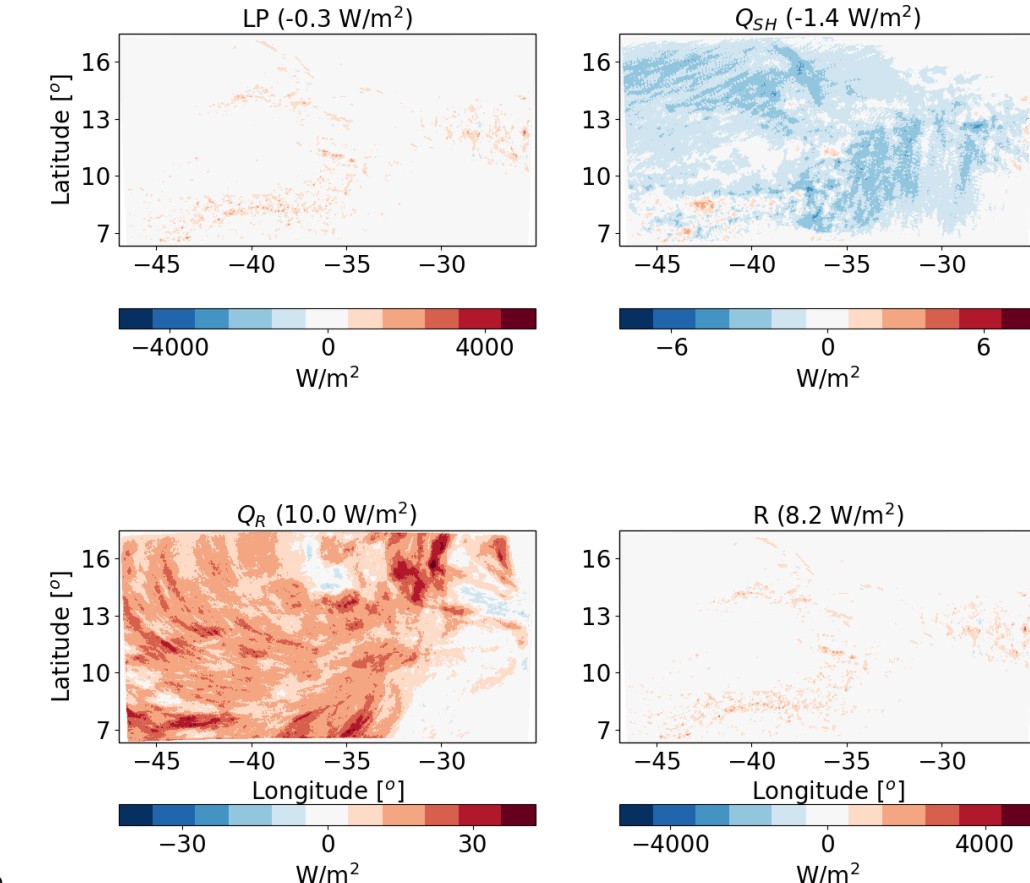


**Figure 14. The differences between polluted (CDNC = 200 cm$^{-3}$) and clean (CDNC = 20 cm$^{-3}$) ICON**

**simulations of the time-mean terms of the energy budget for the deep-cloud dominated case (16-18/08/2016).**

**The terms that appears here are: $LP$ - latent heat by precipitation, $Q_{SH}$ - sensible heat flux, $Q_R$ - atmospheric**

**radiative warming, and R – the residual. The domain and time mean value of each term appears in**

**parenthesis.**


The large increase in $Q_R$ is caused mostly by the increase in $F_{LW}^{TOA}$ (which becomes less negative
i.e. less outgoing LW radiation under polluted conditions – Fig. 15). The CDNC effect on $F_{LW}^{SFC}$
has a much smaller magnitude. The SW fluxes changes are substantial (-14.1 W/m$^2$ at TOA and
-12.3 W/m$^2$ at the surface), however, in terms of the atmospheric energy budget, since clouds do





441 not absorb much in the SW, the TOA and surface changes almost cancel each other out and the

442 net effect is only ~1.8 W/m² atmospheric radiative cooling (which decrease some of the LW

443 warming). The net TOA total (SW+LW) radiative flux change is about -1.9 W/m². The trends in

444 the mean cloud properties (Figs. 16 and 17 below) can explain this large radiative response.

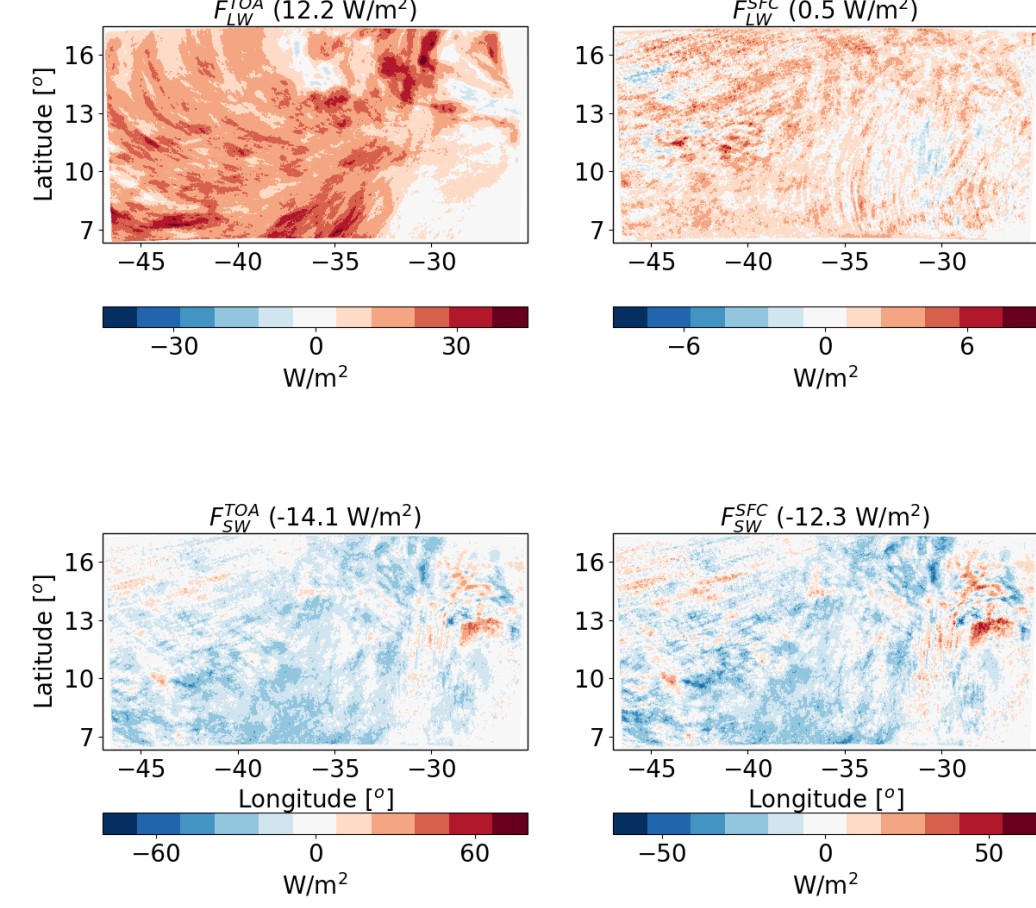

445

446 **Figure 15. The differences between polluted (CDNC = 200 cm⁻³) and clean (CDNC = 20 cm⁻³) ICON**

447 **simulations of the time mean radiative longwave (LW) and shortwave (SW) fluxes at the top of atmosphere**

448 **(TOA) and surface (SFC) for the deep-cloud dominated case (16-18/08/2016). The domain and time mean**

449 **value of each term appears in parenthesis.**

450

451 Figure 16 presents some of the domain mean properties as a function of time for the deep-cloud

452 dominated case. It demonstrates an increase in CF with CDNC which is more significant during





the second day of the simulation. This is opposite to the CF reduction in the shallow-cloud dominated case (Fig. 8). It also demonstrates a very significant increase in LWP and, even more (in relative terms), in IWP and thus also in TWP. The increase in CF and water content can explain the decrease in SW fluxes both at TOA and surface (Fig. 15) as more SW is being reflected back to space. The larger SW reflection under increased CDNC is also contributed to by the Twomey effect (Twomey, 1977). Re-running the simulations without the Twomey effect result in 9.6 W/m$^2$ reduction in the TOA SW flux as compare to 14.1 W/m$^2$ with the Twomey effect on. We note that the relative role of the Twomey effect (compare to the cloud adjustments – CF and TWP) is larger in the shallow-cloud dominated case as compare to the deep-cloud dominated case (-9.6 W/m$^2$ and -14.1 W/m$^2$ for simulations with and without the Twomey effect in the deep-cloud dominated case, compare to -1.7 W/m$^2$ and -7.5 W/m$^2$ in the shallow-cloud dominated case, respectively). However, it should be noted that the Twomey effect due to changes in the ice particles size distribution was not considered. In this case, unlike in the shallow-cloud dominated case, the three contributions to the SW changes (CF, Twomey and LWP/IWP, e.g. (Goren and Rosenfeld, 2014)) all contribute to the SW flux reduction (Fig. 15 presents the results of all contributors). Off-line sensitivity tests demonstrate that the relative contribution of the TWP and the CF to the increase in SW reflectance is roughly ¾ and ¼, respectively.

The vertical profile changes with CDNC (Fig. 17) demonstrate a consistent picture of a decrease in CF in low clouds and a significant increase in CF and liquid and ice content at the mid and upper troposphere. The CF increase at the upper troposphere, and especially the increase in the ice content, can explain the decrease in the outgoing LW radiation (Fig. 15). The increase in ice content at the upper troposphere is in agreement with recent observational studies (Gryspeerdt et al., 2018; Sourdeval et al., 2018; Christensen et al., 2016). Analysis of the upward water mass flux from the warm to the cold part of the clouds (at 500 mb) in the different simulations (Fig. 19), demonstrates a substantial increase with the increase in CDNC (Chen et al., 2017), which occur even without a large change in the vertical velocity due to the increase in the water content (Fig. 17) and the delay in the rain formation to higher levels (Heikenfeld et al., 2019). Similar to the shallow-cloud dominated case (Fig. 8), the near surface temperature monotonically increases with CDNC, while the effect on the mean rain rate is small.

The differences in the thermodynamic evolution between polluted and clean conditions for this case (Fig. 18), demonstrate the same trend as in the shallow-cloud dominated case (Fig. 10). Here again, we note an increase in the humidity at the mid and upper troposphere, that contribute



to the reduction in the outgoing LW flux. The deepening, drying and warming of the boundary
layer are observed in this case as well. Both the increase in humidity at the mid-upper troposphere
and the deepening of the boundary layer (Seifert et al., 2015) could cause a reduction of the
outgoing LW flux. To distinguished the effect of clouds and humidity at the different levels on
the outgoing LW flux, we have conducted sensitivity off-line radiative transfer calculations using
BUGSrad. As in the shallow-cloud dominated case, the difference in outgoing LW flux between
clean and polluted conditions primarily emerges from the CDNC effect on clouds. The small
remaining effect of the clear sky (~0.2 W/m$^2$) is contributed by the change in the humidity at the
mid and upper troposphere rather than by the deepening of the boundary layer (which would lead
to LW emission from lower temperatures and is expected to be more significant under lower free
troposphere humidity conditions).

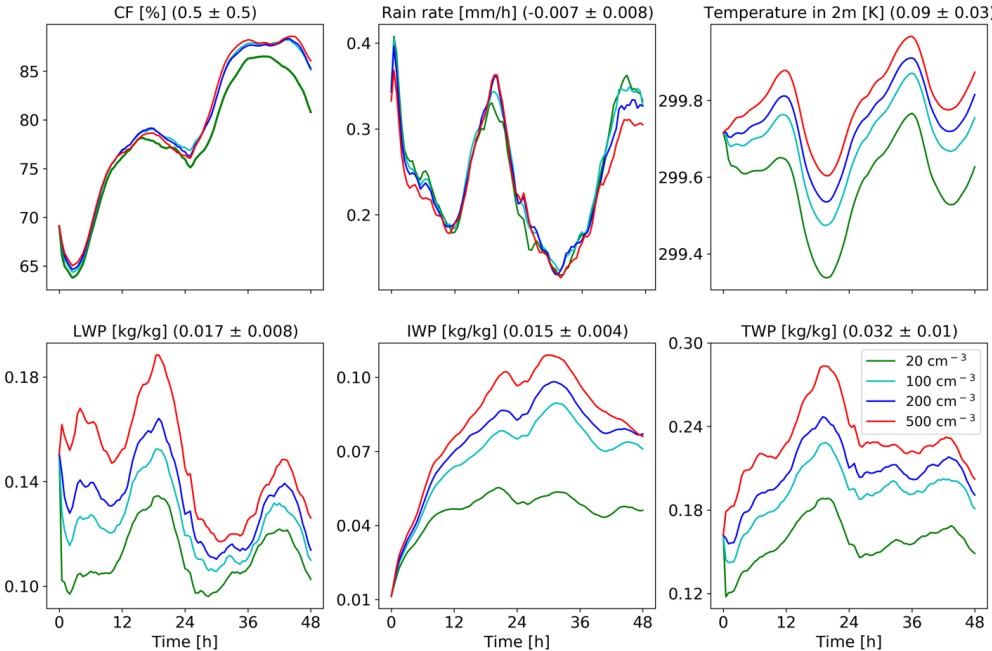


**Figure 16. Domain average properties as a function of time for the different CDNC simulations for the deep-**
**cloud dominated case. The properties that are presented here are: cloud fraction (CF), rain rate, temperature**
**in 2 m, liquid water path (LWP), ice water path (IWP) and total water path (TPW = LWP + IWP). For each**
**property the mean difference between all combinations of simulations, normalized to a factor 5 increase in**
**CDNC, and its standard deviation appear in parenthesis.**


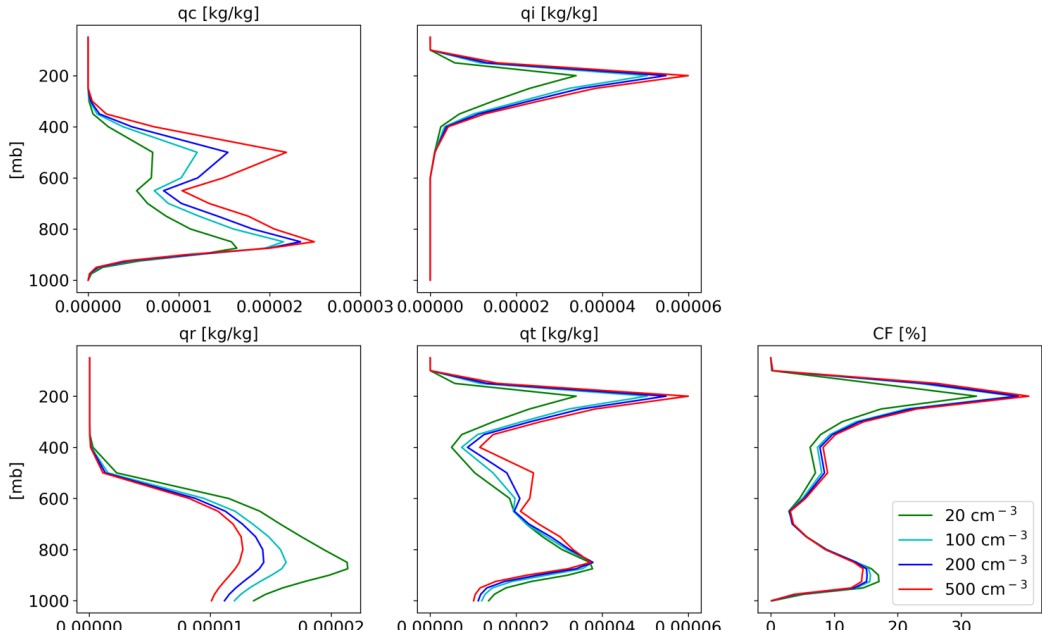


**Figure 17. Domain and time average vertical profiles for the different CDNC simulations for the shallow-cloud dominated case. The properties that are presented here are: cloud droplet mass mixing ratio (qc – for clouds' droplets with radius smaller than 40 μm), ice mass mixing ratio (qi), rain mass mixing ratio (qr - for clouds' drops with radius larger than 40 μm), total water mass mixing ratio (qt = qc+qi+qr), and cloud fraction (CF).**








**Figure 18. Hovmöller diagrams of the differences in the domain mean temperature, specific humidity (qv)**
**and relative humidity (RH) vertical profiles between polluted (CDNC = 200 cm⁻³) and clean (CDNC = 20 cm⁻**
**³) simulations for the deep-cloud dominated case (16-18/08/2016).**

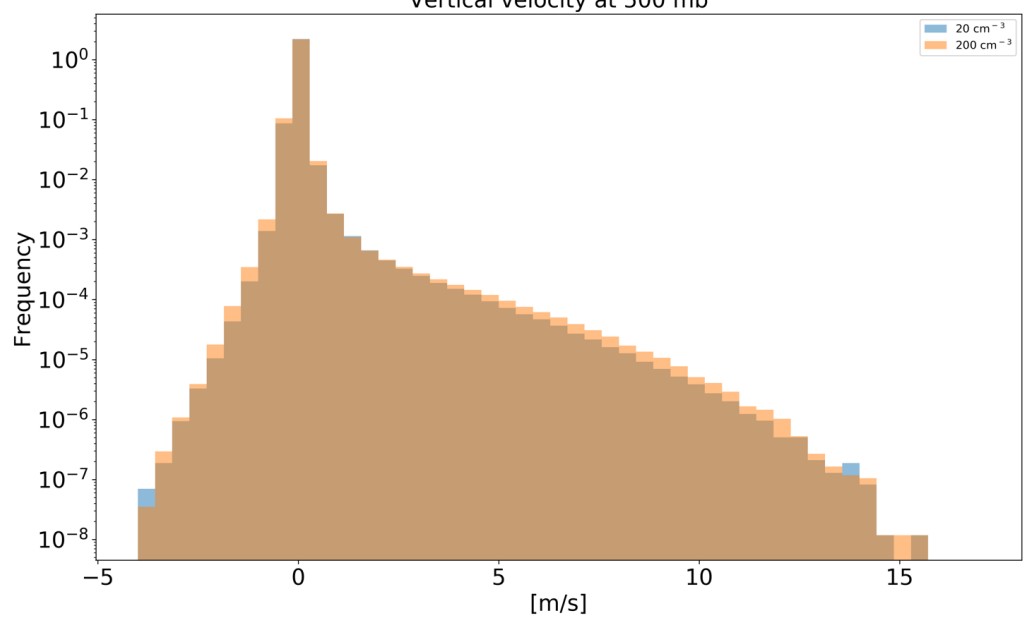

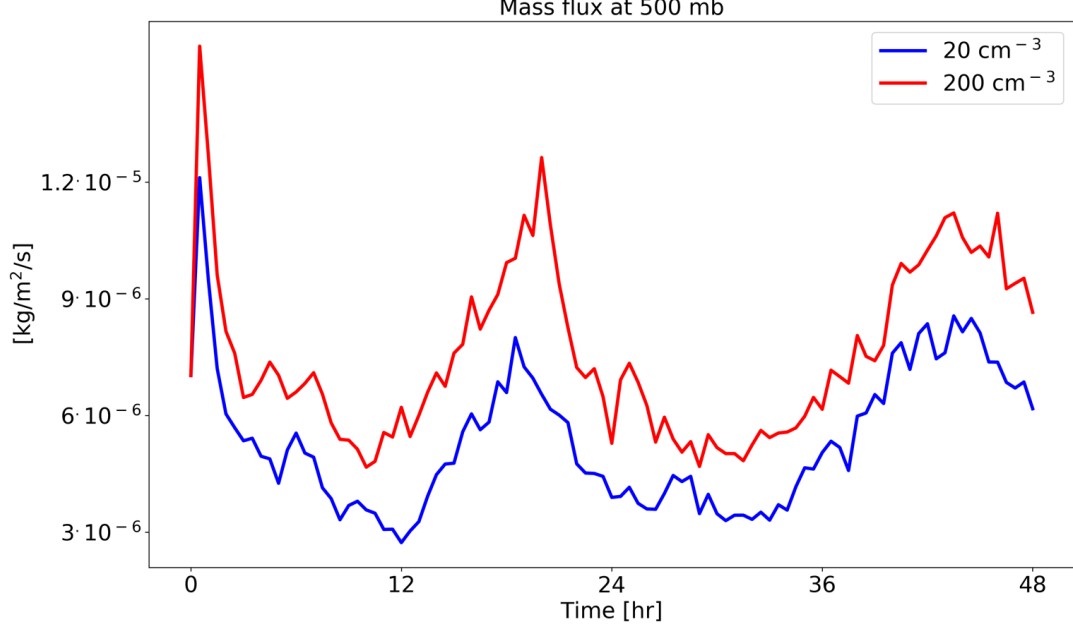






**Figure 19. histograms of ICON simulated vertical velocity at the level of 500 mb (upper), and the time**
**evolution of the net upwards water (liquid and ice) mass flux (lower) for a clean (CDNC = 20 cm⁻³) and**
**polluted (CDNC = 200 cm⁻³) simulations for the deep-cloud dominated case (16-18/08/2016). The 500 mb level**
**is chosen as it represents the transition between the warm part to the cold part of the clouds.**

**Summary and conclusions**
Two different case studies of tropical cloud systems over the Atlantic Ocean were simulated
using the ICON numerical model in a cloud resolving configuration with 1.2 km resolution and
a relatively large domain (~22° x 11°). The cases represent dates from the NARVAL 2 field
campaign that took place during August 2016 and have different dominant cloud types and
different dominating terms in their energy budget. The first case (10-12/8/2016) is shallow-cloud
dominated and hence dominated by radiative cooling, while the second case (16-18/8/2016) is
dominated by deep convective clouds and hence dominated by precipitation warming. The main
objective of this study is to analyse the response of the atmospheric energy budget to changes in
cloud droplet number concentration (CDNC), which serve as a proxy for (or idealized
representation of) changes in aerosol concentration. This enables better understanding of the
processes acting in global-scale studies trying to constrain aerosol effect on precipitation changes
using the energy budget perspective (O'Gorman et al., 2012;Muller and O'Gorman,
2011;Hodnebrog et al., 2016;Samset et al., 2016;Myhre et al., 2017;Liu et al., 2018;Richardson
et al., 2018;Dagan et al., 2019a). Our results demonstrate that regional atmospheric energy
budgets can be significantly perturbed by changes in CDNC and that the magnitude of the effect
is cloud regime dependent (even for a given geographical region and given time of the year as
the two cases are separated by less than a week).
Figure 20 summarizes the energy and radiation response of the two simulated cases to CDNC
perturbations. It shows that the atmosphere in the deep-cloud dominated case experiences a very
strong atmospheric warming due to an increase in CDNC (10.0 W/m²). Most of this warming is
caused by a reduction in the outgoing LW radiation at the TOA. The SW radiative fluxes (both
at the TOA and surface) is also significantly modified but their net effect on the atmospheric
column energy budget is small. The net TOA radiative fluxes change in this case is -1.9 W/m².
Beside the atmospheric radiative warming, changes in precipitation (~-0.3 W/m²), and in sensible
heat flux ($Q_{SH}$, -1.4 W/m²) also contribute to the total trend as a response of increase in CDNC.
We note that since 1 mm/hr of rain is equivalent to 628 W/m², even negligible changes in





precipitation of less than 0.5 mm over 48 hr (as seen in our simulations) can still appear as
significant changes in the atmospheric energy budget and contribute a few W/m$^2$.
The response of the radiative fluxes can be explained by the changes in the mean cloud and
thermodynamic properties in the domain. The mean cloud fraction (CF) increases with the
increase in CDNC (Fig. 16) while the vertical structure of it indicates a reduction in the low
cloud fraction (below 800 mb) and an increase in the mid and upper troposphere CF (Fig. 17).
The water content (both liquid and ice) also increase with the increase in CDNC (Figs. 16 and
17) with increasing amount with height. These changes in the mean cloud properties drive both
the reduction in SW fluxes at TOA and surface and LW flux at TOA as the clouds become more
opaque (Koren et al., 2010; Storelvmo et al., 2011) and cover a larger fraction of the sky. In
addition to cloud responses, the domain-mean thermodynamic conditions change as well (Fig.
18). Specifically, the humidity content at the mid and upper troposphere increases with higher
CDNC, (due to increase mass flux to the upper troposphere) which further decreases the outgoing
LW flux at the TOA. However, the vast majority of the LW effect emerges from the changes in
clouds.
Both the increase in water vapor and ice content at the upper troposphere are driven by an
increase in water mass flux with increasing CDNC to these levels (Fig. 19, (Koren et al., 2005;
Rosenfeld et al., 2008; Altaratz et al., 2014; Chen et al., 2017)), which is caused mostly by the
increase in the water mixing ratio at the mid-troposphere rather than by increase in vertical
velocity (Figs. 11 and 19). The ice content at the upper troposphere is also increased due to
reduction in the ice falling speed (Grabowski and Morrison, 2016), while the increased relative
humidity at these levels, further increases the ice particle lifetime due to slower evaporation.
However, the increase in water mass flux to the upper layers is not accompanied with an increase
in precipitation as predicted by the classical "invigoration" paradigm (Altaratz et al., 2014;
Rosenfeld et al., 2008), which suggest that some compensating mechanisms are operating
(Stevens and Feingold, 2009).
In the shallow-cloud dominated case (which also contains a significant amount of deep
convection), the response of $Q_R$ is weaker but still substantial (a total decrease in the atmospheric
radiative cooling of 1.6 W/m$^2$ - Fig. 20). Here again, the changes in $Q_{SH}$ decrease about -1.4
W/m$^2$ of this atmospheric warming. As in the deep-cloud dominated case, most of the
atmospheric radiative warming is caused by reduction in the outgoing LW flux, while the surface
and TOA SW fluxes changes are non-negligible but cancel each other out (in terms of the





atmospheric energy budget – reflecting small SW atmospheric absorption changes). However, a
significant TOA net (SW+LW) radiative flux change of ~-5.2 W/m$^2$ remains. In this case, the
cloud-mean effect on radiation is more complicated. While CF decreases with increasing CDNC,
the mean water path (both LWP and IWP) increases (Fig. 8). As in the deep-cloud dominated
case, the increase in the water content occurs mostly at the mid and upper troposphere, while the
decrease in CF occurs mostly in the lower troposphere (Fig. 9). In terms of the SW fluxes, the
effect of the decrease in low CF (decrease SW reflections) and the increase in water mass
(increase SW reflections) would partially compensate, while the Twomey effect (Twomey, 1977)
adds to the increase SW reflections. In this case, the net effect is more SW reflected back to
space at TOA and a net negative flux change (including also the LW).
There exists a large spread in estimates of aerosol effects on clouds for different cloud types and
different environmental conditions. In this study, as we use a relatively large domain (22º x 11º)
and two different dates (each for two days), we sample many different local environmental
conditions and cloud types. Such more realistic setups (although with lower spatial resolution)
could provide more reliable estimates of aerosol effects on heterogeneous cloud systems than
just one-cloud-type, small domain simulations (as was done in many previous studies, e.g (Dagan
et al., 2017; Seifert et al., 2015; Ovchinnikov et al., 2014)). In addition, the realistic setup with
the continuously changing boundary conditions and systems that pass through the domain
prevent conclusions that might be valid only in cyclic double periodic large eddy simulations, as
the background meteorological conditions change more realistically (Dagan et al., 2018b).
Another uncertainty in the assessment of the aerosol response are the large differences between
different models and microphysical schemes (White et al., 2017; Fan et al., 2016; Khain et al.,
2015; Heikenfeld et al., 2019). In this study, as we use only one model, we do not address this
uncertainty. In future work we intend to examine the response in multiple models. In addition,
more detailed observational constraints on the models are needed. Furthermore, we do not
include temporal evolution of the aerosol concentration. Feedbacks between the aerosol
concentration and clouds processes (such as wet scavenging) would add another layer of
complexity that should be accounted for in future work.
Generally, the global mean aerosol radiative forcing is estimated to be negative (Boucher et al.,
2013;Bellouin et al., 2019). However, these global aerosol forcing estimates have so far not
included the radiative forcing associated with potential effects of aerosols on deep convection –
and these effects are not represented in most current climate models due to limitations in
convection parameterisations, with only a few exceptions (Kipling et al., 2017; Labbouz et al.,



2018). Here we demonstrate the existence of non-negligible aerosol radiative effects (of -5.2 and
-1.9 W/m$^2$ for the shallow and deep cloud dominated cases, respectively) in tropical cloud
systems, that contained both deep and shallow convective clouds, with significant SW and LW
contributions. From the (limited) two cases simulated here, it appears that (in agreement with
previous studies) the aerosol effect may be regime dependent and that even within a given cloud
regime the effect may vary with the meteorological conditions.
Finally, we hypothesise that the aerosol impact shown on the atmospheric energy balance, with
increasing divergence of dry static energy from deep convective regions concomitantly with
increased convergence in shallow clouds regions, can have effects on the large-scale circulation.
This should be investigated in future work.






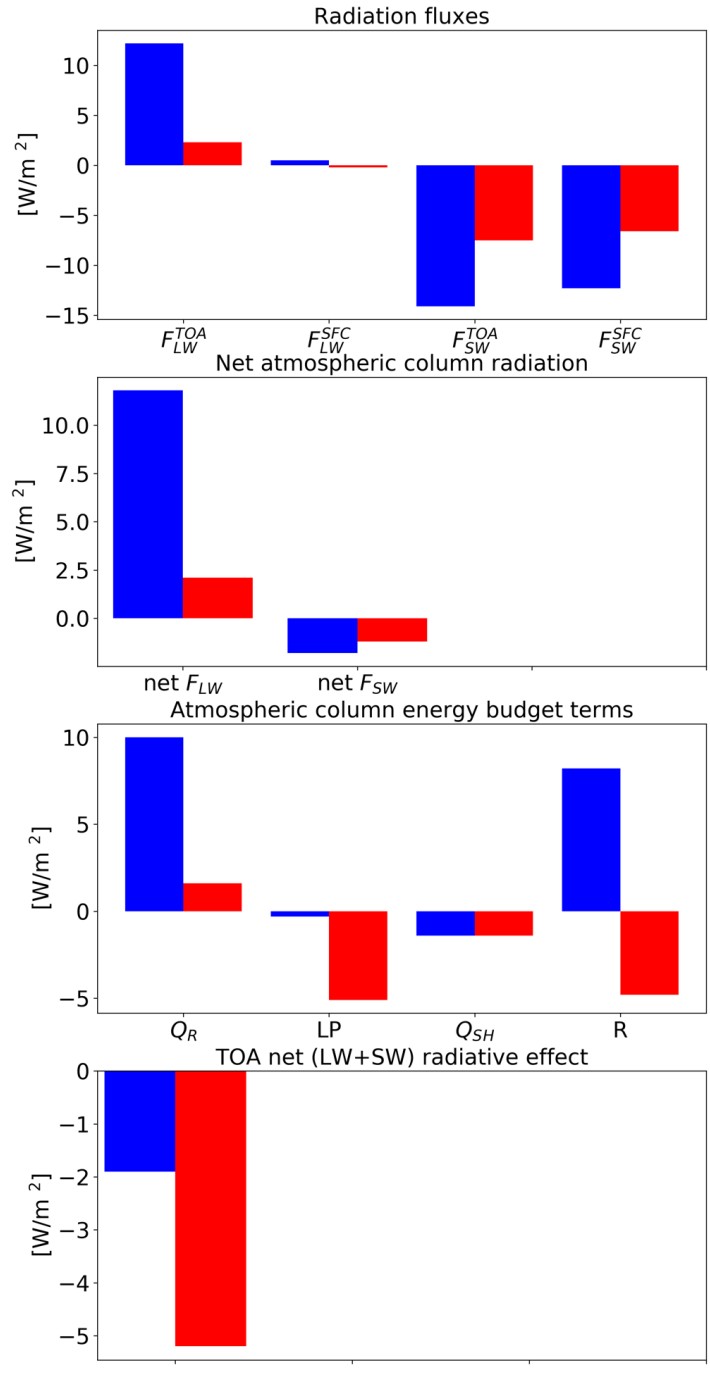

**Figure 20. Summary of the radiation and energy response to CDNC perturbation in the two different cases.**

**Blue represent the deep-cloud dominated case while red the shallow-cloud dominated case.**





**Author contributions.** G. D. carried out the simulations and analyses presented. G.C., D.K. and A.S. assisted with the simulations. M.C. assisted with the radiative transfer calculations and comparison with observations. P. S. and A.S. assisted with the design and interpretation of the analyses. G. D. prepared the manuscript with contributions from all co-authors.

**Acknowledgements:**

This research was supported by the European Research Council (ERC) project constRaining the EffeCts of Aerosols on Precipitation (RECAP) under the European Union's Horizon 2020 research and innovation programme with grant agreement No 724602. The simulations were performed using the ARCHER UK National Supercomputing Service. ECMWF is acknowledged for providing Era-interim data set (https://apps.ecmwf.int/datasets/). We acknowledge MPI, DWD and DKRZ for the NARVAL simulations.

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
