# Peer review of "Atmospheric energy budget response to idealized aerosol perturbation in tropical cloud systems"

_Atmospheric Chemistry and Physics, 2019_

## Referee Comment (RC1) · Anonymous Referee #1 · 3 Jan 2020

The authors have run some fairly large-domain cloud resolving simulations over the tropical Atlantic, in order to investigate how aerosol indirect effects contribute to the radiative impact of aerosols. They note a non-negligible contribution that depends on the cloud regime being examined. The regime and location dependence of aerosol forcing is something that is not well captured in climate models, since most don't include aerosol effects on convection. This study is straightforward and worthwhile, as it begins to break down these differences in the radiation budgets. I believe it is a good contribution and have only a few suggestions to help improve the manuscript.

Comments:

Changing CDNC is, you even admit, a rather simplistic way to approach aerosol effects. In addition to neglecting the activation and scavenging effects that you mention,

another missing piece is the direct effect. This can be especially important in the eastern Atlantic where you are looking since a lot of the aerosols that would be present in this region are dust. Have you considered how the direct effect would fit into this?

You held SST constant in these simulations. Do you have a sense for how much this might have affected the overall energy budgets? I would suspect at least the sensible heat flux might show some differences.

Your 'residual' term is rather large, especially in the deep convective case. You make the point earlier that this term would become negligible on longer time and spatial scales, however an important point in this paper is how large the differences can be on smaller scales. Do you have thoughts on what is largely making up this residual term? How much of it is physical processes that you are not considering, versus the fact that the model simulations are not going to be perfectly balanced, considering the scales and the boundary forcing.

Your mass flux in Fig 11 - how is this calculated? Is this just a total over the whole domain? Or only in updrafts? If this is domain-wide, I imagine the largest reason for the increase is simply the larger amount of deep convection.

The paper is a bit long and I think you could consider getting rid of a few of the figures that tell a redundant story. The inclusion of the 2nd, deep convective case is important because of this point you make on page 32 "Our results demonstrate that regional atmospheric energy budgets can be significantly perturbed by changes in CDNC and that the magnitude of the effect is cloud regime dependent (even for a given geographical region and given time of the year as the two cases are separated by less than a week)." However, the physical mechanisms for the changes in cloud amount and radiative fluxes are consistent between the two cases, so some of the figures and discussion here are a bit repetitive.

In general this could use some copy-editing. Nothing that prevents understanding, but there are a number of small typos and verb agreement issues.

---

## Referee Comment (RC2) · Anonymous Referee #2 · 13 Jan 2020

Review of "Atmospheric energy budget response to idealized aerosol perturbation in tropical cloud systems" by Dagan et al.

This paper studies the differences in the radiative and energy budgets in a tropical environment when the cloud droplet number concentration (CDNC) is changed (as a proxy for changes to the environmental aerosol concentration). The simulated periods are two separate 2-day within the same week where the convection is either predominantly shallow or predominantly deep - each case is simulated with CDNC values. The authors find substantial changes in amount of energy absorbed by the atmosphere by changing the CDNC and find substantial differences in the response between the two sets of simulations.

The main component of the paper is a breakdown of the energy budget into radiative,

sensible heating and heating through precipitation formation. The radiative component is later broken down into shortwave and longwave fluxes at both the surface and top of the atmosphere. The differences are also quantified in the time evolution of near-surface temperature, precipitation, cloud fraction and in-cloud water contents.

Overall, I find the study to be well formulated with a clear motivation and simple but successful strategy for breaking apart the components contributing to the changes in the atmospheric energy budget. There are no substantial shortcomings that should prevent the publication of this study; however, I have a few suggestions that could improve this contribution which are explained below.

Primarily my suggestions are aimed to help the authors achieve their stated aim of better understanding the physical processes behind aerosol effects on the atmospheric energy budget. I see that their study does indeed achieve this, at least partly, but that these results are not clearly expressed in the abstract nor the conclusions. Throughout the paper, the authors do a good job of describing the differences between their simulations and quantifying these differences (although in parts the quantification could be improved) - however, it is mostly left to the reader to put these pieces of information together to get an understanding of the physical processes involved. As a result, my overall impression of the authors conclusions and abstract are: "we found another case where aerosol-cloud interactions behave differently under different environmental conditions," which could be relatively simply converted to "these processes (*see below) contribute to the different energy budget changes for shallow and deep convection when CDNC is changed"

(1) From my understanding of the presented results, it seems that the large difference between the shallow case and the deep case is the potential for a large upper-level cloud fraction change in the deep case. I understand this as an increase of the anvil area, resulting in reduced LW emission from the surface/lower atmosphere and therefore a warming contribution of the larger anvil. In the shallow case, the upper level cloud fraction also has a systematic change, but because it occupies a smaller part of

the model domain - the overall change in the energy budget is controlled by the change in low cloud fraction and the Twomey effect. If the authors agree with this, I suggest adding a paragraph into the conclusions and a sentence in the abstract clarifying these physical changes in the model and their impact on the energy budget.

(2) Breakdown of the vertical mass flux changes with CDNC into component parts The vertical mass flux of water is shown to change between the simulations with different CDCN. What is the cause for this change? Either the vertical velocity should be increasing (which seems not to be the case from the vertical velocity distributions in Figures 11 and 19), so either the updraft area is increasing [implying wider updrafts?] or the in-cloud water mass in increasing [because of a less efficient precipitation-forming processes?]. To what extent are these two factors important? Furthermore, what happens to the vertical mass flux at (e.g.) 800 hPa - where the total water content is quite similar between all CDNC concentrations - is there still an increased vertical mass flux?

(3) Large contribution from residuals A large contribution to the overall energy budget is within the residual term, which the authors state should reduce to zero given a long enough averaging time. How can the authors be sure that this is true and that the large component in the residual term is not a "buffering" effect - e.g. changing stability of the atmosphere to compensate for the changing energy budget? Can the 3D distribution of the residual values be used to quantify this at all?

(4) There appears to be a mismatch between TWP in Figure 8 (lower right plot) and qt in Figure 9 (lower central plot). Similarly in Figures 16 & 17. The vertical profile of qt is quite similar for 3 simulations in the shallow case (Figure 9; excluding the 500 cm-3 line). Similarly, the qt values from 3 simulations in the deep case are also similar (Figure 17; excluding the 20 cm-3 line). However, in figure 8 & 16 these is clear separation between all the TWP lines throughout the simulation. By quick calculation the spread in the TWP timeseries seems too large to be explained by the differences in qt (which are mostly between 650-400 hPa). How can this difference be explained? Is the TWP only including cloud and ice, but ignoring rain water? Similarly, is the LWP

only cloud, ignoring rain?

(5) Following from the above point: is rain water radiatively interactive in the model? If not, to what extent does this removal of mass from the radiatively interactive cloud species have on the Twomey effect calculations performed, given that the rain water mass is almost equal to the cloud water mass at some heights?

(6) Impact of simplifications The approach of simply modifying the CDNC instead of the aerosol concentration of the atmosphere ignores several potentially important processes/feedbacks (e.g. activation of CCN/IN, size distribution of aerosol, direct radiative effects) - the authors should comment on these shortcomings in the conclusions.

(7) Robustness of results The authors should comment on the robustness of these results, in light of the fact that single simulations (rather than ensembles) of two individual case studies are performed. The results in figures 8 & 16 suggest a clear separation between all 4 CDNC concentrations from early on in the simulation - however, the vertical profiles of qi, qt and CF in figures 9 & 17 suggest that the 20 CDNC cm-3 simulation is the only one of the four that is substantially different (particularly at upper levels, which seem to be most important in this story).

Minor points: please show the mass flux from all four simulations in figures 11 and 19 to be consistent with the other plots in the paper.

Lines 335-338: please be more quantitative about the results of the test with the offline radiation calculations as to the relative contributions of the cloud fraction and TWP changes. Line 367: please quantify the "vast majority" of LW flux changes due to cloudy rather than clear skies.

The plots in figures 10 and 18, currently described in the caption as Hovmöller plots, would be better described as time-height plots.

Is there an explanation for the relative minimum of cloud water content at 650 hPa in all simulations? I struggle to find a physical explanation for this.

---

## Author Comment (AC1) · 13 Feb 2020

Response to the reviewers' comments on

**Atmospheric energy budget response to idealized aerosol perturbation in tropical cloud systems**

We would like to thank the revisers for their constructive and thoughtful reviews that helped us improve our paper.

Below please find a point by point reply to all of the reviewers' comments (in blue).

**Reviewer #1:**

The authors have run some fairly large-domain cloud resolving simulations over the tropical Atlantic, in order to investigate how aerosol indirect effects contribute to the radiative impact of aerosols. They note a non-negligible contribution that depends on the cloud regime being examined. The regime and location dependence of aerosol forcing is something that is not well captured in climate models, since most don't include aerosol effects on convection. This study is straightforward and worthwhile, as it begins to break down these differences in the radiation budgets. I believe it is a good contribution and have only a few suggestions to help improve the manuscript.

Reply: we would like to thank the reviewer again for the effort and the constructive comments. We are happy that the reviewer found our paper to be straightforward and worthwhile.

Comments:

Changing CDNC is, you even admit, a rather simplistic way to approach aerosol effects. In addition to neglecting the activation and scavenging effects that you mention, another missing piece is the direct effect. This can be especially important in the eastern Atlantic where you are looking since a lot of the aerosols that would be present in this region are dust. Have you considered how the direct effect would fit into this?

Reply: Thank you for this comment. Indeed, the direct effect of aerosol is interesting and important [and is a main focus of our work, i.e. (Dagan et al., 2019)] but is not included in this study. We believe that separating the overall response of the atmospheric energy budget to the radiative and microphysical aerosol effects is a necessary first step in studying this complex system. This is the approach we are taking in the current study. However, we are currently working on implementing the aerosol model HAM (Stier et al., 2005) into the reginal version of ICON. This will allow studying the mutual interaction between the aerosol radiative and microphysical effects in a cloud and aerosol resolving simulations.

Following this comment, we have added to the revised manuscript the following:

*"The different CDNC scenarios serve as a proxy for different aerosol conditions (as the first order effect of increased aerosol concentration on clouds is to increase the CDNC, Andreae, 2009). This also allows to separate the cloud response from the uncertainties involved in the representation of the aerosols in numerical models (Ghan et al., 2011; Simpson et al., 2014; Rothenberg et al., 2018). However, it limits potential feedbacks between clouds and aerosols, such as the removal of aerosol levels by precipitation scavenging and potential aerosol effects thereon. In addition, the fixed CDNC framework does not capture the differences in aerosol activation between shallow and deep clouds, due to differences in vertical velocity. Another aerosol effect that is not included in our simulations is the direct interaction between aerosol and radiation. In future work we plan to examine the mutual interaction between microphysical effects and the direct aerosol radiative effects."*

In the conclusions:

*"Furthermore, we do not include the temporal evolution of the aerosol concentration. Feedbacks between the aerosol concentration and clouds processes (such as wet scavenging), as well as the direct effects of aerosol on radiation would add another layer of complexity that should be accounted for in future work."*

You held SST constant in these simulations. Do you have a sense for how much this might have affected the overall energy budgets? I would suspect at least the sensible heat flux might show some differences.

Reply: We agree that including interactive SST would predominantly affect the sensible heat flux. However, please note that in both cases the sensible heat flux is an order of magnitude smaller than the rest of the terms (Figs. 4 and 12 in the manuscript). Hence, we do not believe that the effect on the total energy budget would be large. In addition, due to the large heat capacity of the ocean, over two days of simulation the SST is not expected to dramatically change. For example, in the deep-cloud dominated case, the difference in the surface radiative fluxes between the clean and polluted conditions is about 12 W/m². Considering the heat capacity of 50m deep ocean mixed layer results in about 0.005K difference in the SST between the two cases over the two days simulation. In the shallow-cloud dominated case the difference in surface radiative fluxes in about half of the deep-cloud dominated case (6 W/m²), resulting in half the temperature change.

Following this argument, we added a clarification about this point to the revised manuscript:

*"Additional details, such as the surface and atmospheric physics parameterizations, are described in Klocke et al., (2017) and include an interactive surface flux scheme and fixed sea surface temperature (SST). We note that using a fixed SST does not include feedbacks of aerosols on the SST evolution that could change the surface fluxes. However, due to the large heat capacity of the ocean, we do not expect the SST to dramatically change over the two days simulations."*

Your 'residual' term is rather large, especially in the deep convective case. You make the point earlier that this term would become negligible on longer time and spatial scales, however an important point in this paper is how large the differences can be on smaller scales. Do you have thoughts on what is largely making up this residual term? How much of it is physical processes that you are not considering, versus the fact that the model simulations are not going to be perfectly balanced, considering the scales and the boundary forcing.

Reply: What was previously called a "residual" term was changes in the revised manuscript to be refer to as "energy imbalance" as it better describes it. A recent study shows that in order to get close to energy balance the spatial scale should be on the order of ~5000km (Jakob et al., 2019) and the time scale longer than a month (we found a similar scale using GCMs). Our simulations operate on smaller spatial-temporal scales than that and hence it is not surprising that we obtain an imbalance.

The energy imbalance is composed of changes in the storage term and local divergence or convergence of dry-static energy into the domain. In our case, almost the entire imbalance is simply dry static energy that moves in or out of the domain. For example, in the deep-cloud dominated case there is a net production of dry-static energy in the domain by precipitation (which is not entirely balanced by the radiative cooling). This extra dry-static energy is then advected out of the domain.

An explanation about this point was added to the revised manuscript:

*"The total column atmospheric energy budget can be described as follows:*

$$LP + Q_R + Q_{SH} = div(s) + ds/dt \qquad (1)$$

*Equation 1 presents a balance between the latent heating rate (LP - latent heat of condensation [L] times the surface precipitation rate [P]), the surface sensible heat flux ($Q_{SH}$), the atmospheric radiative heating ($Q_R$), the divergence of dry static energy (div(s), which will become negligible on sufficiently large spatial scales), and the dry static energy storage term (ds/dt, which will become negligible on long [inter-annual] temporal scales). Throughout the rest of this paper we*

*will refer to the right-hand side of Equation 1 (div(s)+ds/dt) as the energy imbalance (which is calculated as the residual [R] of the left-hand side)."*

*"In this shallow-cloud dominated case the radiative cooling of the atmosphere is significantly larger than the warming due to precipitation (mean of -114.7 W/m² compared to 90.1 W/m²), hence the energy imbalance (R) is negative. Negative R means that there must be some convergence of dry static energy into the domain and/or decrease in the storage term, in this case it is mostly due to convergence of dry static energy."*

*"Next, we analyse the atmospheric energy budget for the deep-cloud dominated case (Fiona tropical storm – Fig. 12). As opposed to the shallow-cloud dominated case, in this case the LP contribution dominates over the radiative cooling and hence the energy imbalance R is positive and large, suggesting divergence of dry static energy out of the domain."*

Your mass flux in Fig 11 - how is this calculated? Is this just a total over the whole domain? Or only in updrafts? If this is domain-wide, I imagine the largest reason for the increase is simply the larger amount of deep convection.

Reply: Thank you for your comment that helped us clarify this point. Calculating the relative change in the cloud fraction and in the water content between clean and polluted conditions demonstrate the dominate role of the latter in the increase in mass flux. This calculation demonstrates that the cloud fraction (total water content) increases by 20% (72%) at 500mb in the simulation with CDNC=200cm⁻³ compared with the simulations with CDNC=20cm⁻³ in the deep-convection dominated case. Similarity, in the shallow dominated case the cloud fraction (total water content) increases by 22% (85%) at 500mb in the simulation with CDNC=200cm⁻³ compared with the simulations with CDNC=20cm⁻³. These calculations demonstrate that about 80% of the increase in mass flux under polluted conditions occur due to the increase in water content and only about 20% occur due to the increase in cloud fraction (recalling that the vertical velocity is similar between the two simulations). The increase in total water content is caused by warm rain suppression at the lower troposphere.

This is now better explained in the revised manuscript:

*"Both the increase in water vapor and ice content in the upper troposphere are driven by an increase in upward water (liquid and ice) mass flux with increasing CDNC (Fig. 11). An*

*increase in mass flux could be caused by an increase in vertical velocities and/or by an increase in cloud (or updraft) fraction and/or by an increase in cloud water content. In our case, the increases in mass flux is driven partially by the small increase in vertical velocity (especially for updraft between 5 and 10 m/s – Fig. 11), partially by the small increase in cloud faction at this level (Fig. 9) and mostly due to the larger water mass mixing ratio (Fig. 9) that leads to an increase in mass flux even for a given vertical velocity.”*

*“Analysis of the upward water mass flux from the warm to the cold part of the clouds (at 500 mb) in the different simulations (Fig. 19), demonstrates a substantial increase with the increase in CDNC (Chen et al., 2017), which occurs due to the increase in the water content (Fig. 17) and the delay in the rain formation to higher levels (Heikenfeld et al., 2019), even without a large change in the vertical velocity or cloud fraction at this level (Fig.17).”*

The paper is a bit long and I think you could consider getting rid of a few of the figures that tell a redundant story. The inclusion of the 2nd, deep convective case is important because of this point you make on page 32 "Our results demonstrate that regional atmospheric energy budgets can be significantly perturbed by changes in CDNC and that the magnitude of the effect is cloud regime dependent (even for a given geographical region and given time of the year as the two cases are separated by less than a week)." However, the physical mechanisms for the changes in cloud amount and radiative fluxes are consistent between the two cases, so some of the figures and discussion here are a bit repetitive.

Reply: Thank you for this comment. We did consider shortening the paper and not including the figures of the second case but eventually we decided to keep them in as it demonstrate an important point of this paper that the aerosol effect on the atmospheric energy budget is meteorological conditions dependent.

In general this could use some copy-editing. Nothing that prevents understanding, but there are a number of small typos and verb agreement issues.

Reply: Thank you. The manuscript went thought copy editing and corrected accordingly.

**Reviewer #2:**

This paper studies the differences in the radiative and energy budgets in a tropical environment when the cloud droplet number concentration (CDNC) is changed (as a proxy for changes to the environmental aerosol concentration). The simulated periods are two separate 2-day within the same week where the convection is either predominantly shallow or predominantly deep - each case is simulated with CDNC values. The authors find substantial changes in amount of energy absorbed by the atmosphere by changing the CDNC and find substantial differences in the response between the two sets of simulations.

The main component of the paper is a breakdown of the energy budget into radiative, sensible heating and heating through precipitation formation. The radiative component is later broken down into shortwave and longwave fluxes at both the surface and top of the atmosphere. The differences are also quantified in the time evolution of near- surface temperature, precipitation, cloud fraction and in-cloud water contents.

Overall, I find the study to be well formulated with a clear motivation and simple but successful strategy for breaking apart the components contributing to the changes in the atmospheric energy budget. There are no substantial shortcomings that should prevent the publication of this study; however, I have a few suggestions that could improve this contribution which are explained below.

Reply: we would like to thank the reviewer again for the effort and for the suggestions. We are happy that the reviewer found that our paper well formulated.

Primarily my suggestions are aimed to help the authors achieve their stated aim of better understanding the physical processes behind aerosol effects on the atmospheric energy budget. I see that their study does indeed achieve this, at least partly, but that these results are not clearly expressed in the abstract nor the conclusions. Throughout the paper, the authors do a good job of describing the differences between their simulations and quantifying these differences (although in parts the quantification could be improved) - however, it is mostly left to the reader to put these pieces of information together to get an understanding of the physical processes involved. As a result, my overall impression of the authors conclusions and abstract are: "we found another case where aerosol-cloud interactions behave differently under different environmental conditions," which could be relatively simply converted to "these processes (*see below) contribute to the different energy budget changes for shallow and deep convection when CDNC is changed"

(1) From my understanding of the presented results, it seems that the large difference between the shallow case and the deep case is the potential for a large upper-level cloud fraction change in the deep case. I understand this as an increase of the anvil area, resulting in reduced LW emission from the surface/lower atmosphere and therefore a warming contribution of the larger anvil. In the shallow case, the upper level cloud fraction also has a systematic change, but because it occupies a smaller part of the model domain - the overall change in the energy budget is controlled by the change in low cloud fraction and the Twomey effect. If the authors agree with this, I suggest adding a paragraph into the conclusions and a sentence in the abstract clarifying these physical changes in the model and their impact on the energy budget.

Reply: Thank you for this suggestion that help us clarify this point. The reviewer's point is a main conclusion of our paper; however, we believe that our results present more than just that and include (among other) the effect of the thermodynamic evolution on radiation under different CDNC conditions, the effect of CDNC on surface fluxes and so on. Nevertheless, following the reviewer's comment, we have added a clarification to the revised manuscript.

In the abstract:

*"It is shown that the total column atmospheric radiative cooling is substantially reduced with CDNC in the deep-cloud dominated case (by ~10.0 W/m$^2$), while a much smaller reduction (~1.6 W/m$^2$) is shown in the shallow-cloud dominated case. This trend is caused by an increase in the ice and water vapor content at the upper troposphere that leads to a reduced outgoing longwave radiation, an effect which is stronger under deep-cloud dominated conditions."*

In the conclusions section:

*"Both the increase in water vapor and ice content in the upper troposphere are driven by an increase in water mass flux with increasing CDNC to these levels (Fig. 19, (Koren et al., 2005; Rosenfeld et al., 2008; Altaratz et al., 2014; Chen et al., 2017)), which is caused mostly by the increase in the water mixing ratio in the mid-troposphere rather than by increase in vertical velocity (Fig. 19) or in cloud fraction (Fig. 17). The ice content in the upper troposphere is also increased due to reduction in the ice falling speed (Grabowski and Morrison, 2016), while the increased relative humidity at these levels, further increases the ice particle lifetime due to slower evaporation."*

*"In the shallow-cloud dominated case (which also contains a significant amount of deep convection), the response of $Q_R$ is weaker but still substantial (a total decrease in the atmospheric radiative cooling of 1.6 W/m$^2$ - Fig. 20). The weaker total response under the shallow-cloud dominated conditions is due to the smaller role of the ice part in this case."*

(2) Breakdown of the vertical mass flux changes with CDNC into component parts
The vertical mass flux of water is shown to change between the simulations with different
CDCN. What is the cause for this change? Either the vertical velocity should be increasing
(which seems not to be the case from the vertical velocity distributions in Figures 11 and 19),
so either the updraft area is increasing [implying wider updrafts?] or the in-cloud water mass
in increasing [because of a less efficient precipitation-forming processes?]. To what extent are
these two factors important? Furthermore, what happens to the vertical mass flux at (e.g.) 800
hPa - where the total water content is quite similar between all CDNC concentrations - is there
still an increased vertical mass flux?

Reply: Thank you for this comment that helped us clarify this point. Calculating the relative
change in the cloud fraction and in the water content between clean and polluted conditions
demonstrate the dominate role of the later in the increase in mass flux. This calculation
demonstrates that the cloud fraction (total water content) increases by 20% (72%) at 500mb in
the simulation with CDNC=200cm$^{-3}$ compared with the simulations with CDNC=20cm$^{-3}$ in the
deep convection dominated case. Similarity, in the shallow dominated case the cloud fraction
(total water content) increases by 22% (85%) at 500mb in the simulation with CDNC=200cm$^{-3}$
compared with the simulations with CDNC=20cm$^{-3}$. These calculations demonstrate that about
80% of the increase in mass flux under polluted conditions occur due to the increase in water
content and only about 20% occur due to the increase in cloud fraction (recalling that the
vertical velocity is similar between the two simulations). The increase in total water content is
caused by warm rain suppuration at the lower troposphere.

This is now better explained in the revised manuscript:

*"Both the increase in water vapor and ice content in the upper troposphere are driven by an
increase in upward water (liquid and ice) mass flux with increasing CDNC (Fig. 11). An
increase in mass flux could be caused by an increase in vertical velocities and/or by an increase
in cloud (or updraft) fraction and/or by an increase in cloud water content. In our case, the
increases in mass flux is driven partially by the small increase in vertical velocity (especially
for updraft between 5 and 10 m/s – Fig. 11), partially by the small increase in cloud faction at
this level (Fig. 9) and mostly due to the larger water mass mixing ratio (Fig. 9) that leads to
an increase in mass flux even for a given vertical velocity."*

*"Analysis of the upward water mass flux from the warm to the cold part of the clouds (at 500
mb) in the different simulations (Fig. 19), demonstrates a substantial increase with the increase*

*in CDNC (Chen et al., 2017), which occur due to the increase in the water content (Fig. 17) and the delay in the rain formation to higher levels (Heikenfeld et al., 2019), even without a large change in the vertical velocity or cloud fraction at this level (Fig.17)."*

In addition, we calculated the mass flux at the level of 800 mb as the reviewer suggested (Figs. R1 and R2 below). It demonstrates that in this level there is also a general increase in mass flux with the increase in CDNC but to a lesser extent. The increase in mass flux is driven by a small increase in water content (Figs. 9 and 17) and a small increase in vertical velocity (Figs. R1 and R2), while the cloud fraction is similar between the simulations (Figs. 9 and 17). In the manuscript we still present the mass flux at 500 mb as it represents the transition between the warm and the cold parts of the clouds.

[Figure]

**Figure R1.** histograms of ICON simulated vertical velocity at the level of 800 mb for a clean (CDNC = 20 cm⁻³) and polluted (CDNC = 200 cm⁻³) simulations (upper), and the time evolution of the net upwards water (liquid and ice) mass flux (lower) for the different CDNC simulations for the shallow-cloud dominated case (10-12/08/2016). In the histogram only two simulations are presented for clarity.

[Figure]

**Figure R2. histograms of ICON simulated vertical velocity at the level of 800 mb for a clean (CDNC = 20 cm⁻³) and polluted (CDNC = 200 cm⁻³) simulations (upper), and the time evolution of the net upwards water (liquid and ice) mass flux (lower) for the different CDNC simulations for the deep-cloud dominated case (16-18/08/2016). In the histogram only two simulations are presented for clarity.**

(3) Large contribution from residuals

A large contribution to the overall energy budget is within the residual term, which the authors state should reduce to zero given a long enough averaging time. How can the authors be sure

that this is true and that the large component in the residual term is not a "buffering" effect - e.g. changing stability of the atmosphere to compensate for the changing energy budget? Can the 3D distribution of the residual values be used to quantify this at all?

Reply: A similar argument was raised by reviewer #1 – we will repeat our reply here. What was previously called a "residual" term was changes in the revised manuscript to be refer to as "energy imbalance" as it better describes it. A recent study shows that in order to get close to energy balance the spatial scale should be on the order of ~5000km (Jakob et al., 2019) and the time scale longer than a month (we found a similar scale using GCMs). Our simulations operate on smaller spatial-temporal scales than that and hence it is not surprising that we obtain an imbalance.

The energy imbalance is composed of changes in the storage term and local divergence or convergence of dry-static energy into the domain. In our case, almost the entire imbalance is simply dry static energy that moves in or out of the domain. For example, in the deep-cloud dominated case there is a net production of dry-static energy in the domain by precipitation (which is not entirely balanced by the radiative cooling). This extra dry-static energy is then advected out of the domain.

An explanation about this point was added to the revised manuscript:

*"The total column atmospheric energy budget can be described as follows:*

$$LP + Q_R + Q_{SH} = div(s) + ds/dt \qquad (1)$$

*Equation 1 presents a balance between the latent heating rate (LP - latent heat of condensation [L] times the surface precipitation rate [P]), the surface sensible heat flux ($Q_{SH}$), the atmospheric radiative heating ($Q_R$), the divergence of dry static energy (div(s), which will become negligible on sufficiently large spatial scales), and the dry static energy storage term (ds/dt, which will become negligible on long [inter-annual] temporal scales). Throughout the rest of this paper we will refer to the right-hand side of Equation 1 (div(s)+ds/dt) as the energy imbalance (which is calculated as the residual [R] of the left-hand side)."*

*"In this shallow-cloud dominated case the radiative cooling of the atmosphere is significantly larger than the warming due to precipitation (mean of -114.7 W/m² compared to 90.1 W/m²), hence the energy imbalance (R) is negative. Negative R means that there must be some convergence of dry static energy into the domain and/or decrease in the storage term, in this case it is mostly due to convergence of dry static energy."*

*"Next, we analyse the atmospheric energy budget for the deep-cloud dominated case (Fiona tropical storm – Fig. 12). As opposed to the shallow-cloud dominated case, in this case the LP contribution dominates over the radiative cooling and hence the energy imbalance R is positive and large, suggesting divergence of dry static energy out of the domain."*

(4) There appears to be a mismatch between TWP in Figure 8 (lower right plot) and qt in Figure 9 (lower central plot). Similarly in Figures 16 & 17. The vertical profile of qt is quite similar for 3 simulations in the shallow case (Figure 9; excluding the 500 cm-3 line). Similarly, the qt values from 3 simulations in the deep case are also similar (Figure 17; excluding the 20 cm-3 line). However, in figure 8 & 16 these is clear separation between all the TWP lines throughout the simulation. By quick calculation the spread in the TWP timeseries seems too large to be explained by the differences in qt (which are mostly between 650-400 hPa). How can this difference be explained? Is the TWP only including cloud and ice, but ignoring rain water? Similarly, is the LW only cloud, ignoring rain?

Reply: Thank you for this comment that helped us clarify this point. Indeed, the LWP include the cloud mass (qc) and not the rain mass (qr). This is done for consistency with LWP calculated from satellite observations, which are sensitive only to the cloud mass and not to the rain mass (see also the reply to the next point). This is now better explained in the revised manuscript:

*"**Figure 8. Domain average properties as a function of time for the different CDNC simulations for the shallow-cloud dominated case. The properties that are presented here are: cloud fraction (CF), rain rate, temperature in 2 m, liquid water path (LWP – based on the cloud water mass, excluding the rain mass for consistency with satellite observations), ice water path (IWP) and total water path (TPW = LWP + IWP). For each property, the mean difference between all combinations of simulations, normalized to a factor 5 increase in CDNC, and its standard deviation appear in parenthesis.**"*

(5) Following from the above point: is rain water radiatively interactive in the model? If not, to what extent does this removal of mass from the radiatively interactive cloud species have on the Twomey effect calculations performed, given that the rain water mass is almost equal to the cloud water mass at some heights?

Reply: As done in most atmospheric models (Hill et al., 2018), the rain mass is not included in the radiative calculations. Since the rain drops are much larger than the cloud droplet, their cross-section available for interaction with radiation (for a given water mass) is much smaller

and usually negligible (Hill et al., 2018). For example, a simple "back of the envelop" calculation of the cloud optical depth ($\tau$) as in Heus & Seifert (2013) and Spill et al. (2019) follows as:

$\tau = 0.19 * LWP^{5/6} * N^{1/3}$,

where LWP is the liquid water path and N is the drop concentration, for a given LWP for cloud droplets with radius r=10 μm and for rain drops with r=0.5 mm yield a factor of 50 decrease in $\tau$ for the rain compare with the cloud. Generally, as $\tau$ is proportional to $N^{1/3}$ it decreases proportional to r (for a given LWP), and since the cloud droplets and rain drops are separated by 1 or 2 orders of magnitudes, the effect of the cloud droplets on the radiation is much larger than that of the rain drops. This simple calculation does not account for the changes in Mie size parameter between rain drops and cloud droplets but it serves to demonstrate the orders of magnitude differences between the two different regimes.

(6) Impact of simplifications

The approach of simply modifying the CDNC instead of the aerosol concentration of the atmosphere ignores several potentially important processes/feedbacks (e.g. activation of CCN/IN, size distribution of aerosol, direct radiative effects) - the authors should comment on these shortcomings in the conclusions.

Reply: Thank you. Based on this comment we have added a comment about the limitation of using a fixed CDNC simulations:

*"The different CDNC scenarios serve as a proxy for different aerosol conditions (as the first order effect of increased aerosol concentration on clouds is to increase the CDNC, Andreae, 2009). This also allows to separate the cloud response from the uncertainties involved in the representation of the aerosols in numerical models (Ghan et al., 2011; Simpson et al., 2014; Rothenberg et al., 2018). However, it limits potential feedbacks between clouds and aerosols, such as the removal of aerosol levels by precipitation scavenging and potential aerosol effects thereon. In addition, the fixed CDNC framework does not capture the differences in aerosol activation between shallow and deep clouds, due to differences in vertical velocity. Another aerosol effect that is not included in our simulations is the direct interaction between aerosol and radiation. In future work we plan to examine the mutual interaction between microphysical effects and the direct aerosol radiative effects."*

In the conclusions:

*"Furthermore, we do not include the temporal evolution of the aerosol concentration. Feedbacks between the aerosol concentration and clouds processes (such as wet scavenging), as well as the direct effects of aerosol on radiation would add another layer of complexity that should be accounted for in future work."*

(7) Robustness of results

The authors should comment on the robustness of these results, in light of the fact that single simulations (rather than ensembles) of two individual case studies are performed. The results in figures 8 & 16 suggest a clear separation between all 4 CDNC concentrations from early on in the simulation - however, the vertical profiles of qi, qt and CF in figures 9 & 17 suggest that the 20 CDNC cm-3 simulation is the only one of the four that is substantially different (particularly at upper levels, which seem to be most important in this story).

Reply: The robustness of our simulations, which, as the reviewer stated, are based on few simulations rather then on large ensemble, occupied our mind as well. In a recent paper which is currently under discussion in ACPD (Dagan and Stier, 2019) we investigate this exact question. In that paper we use a smaller domain compared to the current study ($3^o$ x $3^o$ rather than $22^o$ x $11^o$) to simulated a large ensemble of initial conditions (all together we simulate 124 different simulations). This large ensemble enables robust identification of the effect of CDNC changes on cloud properties. We were able to show that the general conclusions which are stated in the current study hold also for large statistics. Based on the following comment we have added a discussion about the robustness of our results to the revised manuscript:

*"There exists a large spread in estimates of aerosol effects on clouds for different cloud types and different environmental conditions. In this study, as we use a relatively large domain ($22^o$ x $11^o$) and two different dates (each for two days), we sample many different local environmental conditions and cloud types. Such more realistic setups (although with lower spatial resolution) could provide more reliable estimates of aerosol effects on heterogeneous cloud systems than just one-cloud-type, small domain simulations (as was done in many previous studies, e.g (Dagan et al., 2017; Seifert et al., 2015; Ovchinnikov et al., 2014)). However, the conclusions demonstrated here are based on two specific cases. In order to examine the validity of our main conclusions over a wider range of initial conditions, we have conducted a large ensemble of simulations starting from realistic initial conditions (although with a smaller domain) in a companion paper (Dagan and Stier, 2019). These simulations*

*demonstrate that the main conclusions presented in this paper are robust and hold also for a wide range of initial conditions representative for this area."*

Minor points: please show the mass flux from all four simulations in figures 11 and 19 to be consistent with the other plots in the paper.

Reply: The mass flux of all four simulations are now presented in Figs. 11 and 19:

[Figure]

**Figure 11. histograms of ICON simulated vertical velocity at the level of 500 mb for a clean (CDNC = 20 cm⁻³) and polluted (CDNC = 200 cm⁻³) simulations (upper), and the time evolution of the net upwards water (liquid and ice) mass flux (lower) for the different CDNC simulations for the shallow-cloud dominated case (10-12/08/2016). The 500 mb level is chosen as it represents the transition between the warm part to the cold part of the clouds. In the histogram only two simulations are presented for clarity.**

[Figure]

**Figure 19. histograms of ICON simulated vertical velocity at the level of 500 mb for a clean (CDNC = 20 cm⁻³) and polluted (CDNC = 200 cm⁻³) simulations (upper), and the time evolution of the net upwards water (liquid and ice) mass flux (lower) for the different CDNC simulations for the deep-cloud dominated case (16-18/08/2016). The 500 mb level is chosen as it represents the transition between the warm part to the cold part of the clouds. In the histogram only two simulations are presented for clarity.**

Lines 335-338: please be more quantitative about the results of the test with the offline radiation calculations as to the relative contributions of the cloud fraction and TWP changes. Line 367: please quantify the "vast majority" of LW flux changes due to cloudy rather than clear skies.
Reply: Thank you. More information was added to the revised manuscript:
*"For estimating the relative contribution of the changes in CF and water content to the SW flux changes we have conducted off-line radiative transfer sensitivity tests. To quantify the water content radiative effect, we feed the same CF vertical profile from the model into the offline radiative transfer model BUGSrad, while allowing the water content vertical profile to change (and visa versa to compute the CF radiative effect). This approach demonstrates that the contribution from the small reduction in CF is negligible compared to the increased SW reflectance caused by the increased water content (the effect of the reduction in CF compensate only about 1% of the effect of the increase in the water content)."*

*"The increased humidity at the upper troposphere would act to decrease the outgoing LW flux, similar to the effect of the increased ice content in the upper troposphere (Fig. 9). However, sensitivity studies with off-line radiative transfer calculations using BUGSrad demonstrate that the vast majority (more than 99%) of the different in $F_{LW}^{TOA}$ between clean and polluted conditions emerges from the cloudy skies (rather than clear-sky), suggesting that the effect of the increased ice content at the upper troposphere dominates."*

The plots in figures 10 and 18, currently described in the caption as Hovmöller plots, would be better described as time-height plots.
Reply: Thanks. It was changed according to the reviewer's suggestion.

Is there an explanation for the relative minimum of cloud water content at 650 hPa in all simulations? I struggle to find a physical explanation for this.

Reply: We think that the relative minimum of domain-wide cloud water at 650 hPa is simply due to the fact that below this level there are still quite a lot of shallow clouds while above it there are anvils clouds with longer lifetime (and hence the larger cloud water mass) then the liquid at lower levels.

**References**

Dagan, G. and Stier, P.: Ensemble daily simulations for elucidating cloud–aerosol interactions under a large spread of realistic environmental conditions, Atmos. Chem. Phys. Discuss., https://doi.org/10.5194/acp-2019-949, in review, 2019.

Dagan, G., Stier, P., & Watson-Parris, D. (2019). Contrasting response of precipitation to aerosol perturbation in the tropics and extra-tropics explained by energy budget considerations. Geophysical research letters.

Heus, T., & Seifert, A. (2013). Automated tracking of shallow cumulus clouds in large domain, long duration large eddy simulations. Geoscientific Model Development, 6(4), 1261-1273.

Hill, P., Chiu, J., Allan, R., & Chern, J. D. (2018). Characterizing the radiative effect of rain using a global ensemble of cloud resolving simulations. Journal of Advances in Modeling Earth Systems, 10(10), 2453-2470.

Jakob, C., Singh, M., & Jungandreas, L. (2019). Radiative Convective Equilibrium and Organized Convection: An Observational Perspective. Journal of Geophysical Research: Atmospheres, 124(10), 5418-5430.

Spill, G., Stier, P., Field, P. R., & Dagan, G. (2019). Effects of aerosol in simulations of realistic shallow cumulus cloud fields in a large domain. Atmospheric Chemistry and Physics.

Stier, P., Feichter, J., Kinne, S., Kloster, S., Vignati, E., Wilson, J., et al. (2005). The aerosol-climate model ECHAM5-HAM. Atmospheric Chemistry and Physics, 5(4), 1125-1156.

---

## Author Response (AR2)

Response to the editor's comment on

**Atmospheric energy budget response to idealized aerosol perturbation in tropical cloud systems**

We would like to thank the editor for handling our paper and for the constructive comment. Below please find a reply to the editors' comment (in blue).

Thank you for your thorough response to the equally thorough reviews. Through the interactive dynamics of the problem over regional scales, your paper provides a level of sophistication and detail to the problem of aerosol-cloud interactions in tropical regions that I believe to be a significant advance.

There are of course aspects of the problem that remain to be considered as you acknowledge in the conclusions and in lines 684 to 690. Before accepting the article for publication I would like to see some statement in the abstract to the effect of your concluding statement that "Furthermore, we do not include the temporal evolution of the aerosol concentration. Feedbacks between the aerosol concentration and clouds processes (such as wet scavenging), as well as the direct effects of aerosol on radiation would add another layer of complexity that should be accounted for in future work." It is notable that the strongest impact you find at the TOA is with convective clouds that would presumably be most efficient at reducing aerosol concentrations through rainout.

**Reply:** Thank you again for this comment. We have added this point to the abstract:

[revised manuscript text omitted]